# The effect of calcaneus and metatarsal head offloading insoles on healthy subjects' gait kinematics, kinetics, asymmetry, and the implications for plantar pressure management: A pilot study

Jiawei Shuang[1], Athia Haron[1], Garry Massey[2], Maedeh Mansoubi[2], Helen Dawes[2], Frank L. Bowling[3,4], Neil D. Reeves[5,6], Andrew Weightman[1], Glen Cooper[1] *

1 School of Engineering, University of Manchester, Manchester, United Kingdom, 2 NIHR Exeter BRC, Exeter, United Kingdom, 3 Faculty of Biology, School of Medical Sciences, Medicine and Health, University of Manchester, Manchester, United Kingdom, 4 Manchester Academic Health Science Centre, Manchester University NHS Foundation Trust, Manchester, United Kingdom, 5 Faculty of Science and Engineering, Department of Life Sciences, Manchester Metropolitan University, Manchester, United Kingdom, 6 Manchester Metropolitan University Institute of Sport, Manchester, United Kingdom

* glen.cooper@manchester.ac.uk

**Data Availability Statement:** All S1 Unprocessed data of plantar pressure, ground reaction force and

## Abstract

### Background

The global number of people with diabetes is estimated to reach 643 million by 2030 of whom 19–34% will present with diabetic foot ulceration. Insoles which offload high-risk ulcerative regions on the foot, by removing insole material, are the main contemporary conservative treatment to maintain mobility and reduce the likelihood of ulceration. However, their effect on the rest of the foot and relationship with key gait propulsive and balance kinematics and kinetics has not been well researched.

### Purpose

The aim of this study is to investigate the effect of offloading insoles on gait kinematics, kinetics, and plantar pressure throughout the gait cycle.

### Methods

10 healthy subjects were recruited for this experiment to walk in 6 different insole conditions. Subjects walked at three speeds on a treadmill for 10 minutes while both plantar pressure and gait kinematics, kinetics were measured using an in-shoe pressure measurement insole and motion capture system/force plates. Average peak plantar pressure, pressure time integrals, gait kinematics and centre of force were analysed.

### Results

The average peak plantar pressure and pressure time integrals changed by -30% (-68% to 3%) and -36% (-75% to -1%) at the region of interest when applying offloading insoles,

gait kinematics files are available from the Mendeley data database (DOI: 10.17632/hfz543gjc7.1).

**Funding:** This work was partially funded by Engineering and Physical Sciences Research Council (EPSRC) (grant number EP/W00366X/1 to A.W. and H.D.). H.D. and M.M. are supported by NIHR Exeter BRC.

**Competing interests:** The authors have declared that no competing interests exist.

whereas the heel strike and toe-off velocity changed by 15% (-6% to 32%) and 12% (-2% to 19%) whilst walking at three speeds.

## Conclusion

The study found that offloading insoles reduced plantar pressure in the region of interest with loading transferred to surrounding regions increasing the risk of higher pressure time integrals in these locations. Heel strike and toe-off velocities were increased under certain configurations of offloading insoles which may explain the higher plantar pressures and supporting the potential of integrating kinematic gait variables within a more optimal therapeutic approach. However, there was inter-individual variability in responses for all variables measured supporting individualised prescription.

## Introduction

The number of people with diabetes increased from 151 million to 537 million between 2000 and 2021 and it is estimated to reach 643 million by 2030 if no effective measures are taken [1]. 6.7 million deaths in 2021 are directly attributed to diabetes around the world [1]. One of the most prevalent and serious diabetic complications worldwide is diabetic foot ulceration (DFU) [2]. 19–34% of diabetics will present with DFU at some point in their lifetime [3]. Amputation is the most expensive and terrifying outcome for diabetic patients, and their risk is 10 to 30 times higher than that of the general population [4]. The total cost associated with diabetes is estimated to reach 1.03 trillion US dollars around the world by 2030 [1]. Researchers and clinicians agree that prevention of DFU is a better strategy than treatment both economically and for patient outcomes [5].

General interventions for diabetic foot ulcer (DFU) include self-care practices, education and self-management, the employment of footwear and orthotic devices, along with diverse clinical treatment strategies [6]. Among these, the use of appropriate footwear stands out as a critical element in the prevention of DFU [6]. Specially designed insoles can be part of an effective DFU prevention strategy which includes maintaining mobility and physical activity as part of a holistic management package [7,8]. These include insoles which offload the high-risk region of the foot, reducing plantar pressure and therefore the risk of DFU [9,10]. Significantly higher peak forefoot pressures are observed in diabetic subjects, for example researchers report 608 kPa and 373 kPa in diabetics with severe and moderate neuropathic, compared to294 kPa and 323 kPa in the mild neuropathic and nonneuropathic diabetic groups (P < 0.0001), suggesting an important role in the etiology of the diabetic foot condition [11]. Lin et al. [9] evaluated the average peak plantar pressure in the forefoot region of an insole with removable plugs and a support arch using a pedar in-shoe plantar pressure measurement system on diabetic patients, and the result demonstrated that the insole can reduce the average peak plantar pressure by up to 41.8% through removing the plugs and adding an arch. Chanda et al. [12] designed a novel custom offloading insole using a finite element model considering the insole material and offloading aperture shape which resulted in a maximum 91.5% decrease of peak von Mises stress at the ulcer region. Although offloading insoles can effectively decrease pressure at high-risk regions, it is important to consider the pressure increase surrounding the offloading area. Penny et al. [10] measured normal plantar pressure of healthy subjects wearing the PegAssist insole system (Darco International, US) and FORS-15 offloading insole (Saluber,

Italy) using F-Scan in-shoe pressure measuring system (TekScan, US), and their results demonstrated that the pixelated insole can offload a forefoot wound by up to 40.1%. Penny et al. [10] found the pressure increased around the offloading aperture edge was found but they did not investigate the pressure in detail. When applying aperture below the high-risk region, the normal pressure at the region periphery will increase, which is known as "edge-effect" [13]. Although the measurements were conducted on healthy subject, it showed the feasibility of protecting high risk region using offloading insoles [10]. Shaulian et al. [14] tested a graded stiffness offloading insole on healthy subjects and the result showed that the insole could improve edge effect problems. In addition, Shaulian et al. [2] developed a novel finite element method to reduce the heel load considering the pressure increase around the offloading aperture edge and two offloading aperture shapes observed the heel load minimization which are both with large offloading radius and depth.

Previous research has clearly shown that offloading insoles are effective at reducing normal plantar pressure in a specific region, with some impact on surrounding areas, however, their potential effect on critical gait kinematics has not received adequate attention. People with diabetes have altered sensorimotor function affecting safe and efficient gait propulsion and balance [15,16]. The effect of standard offloading insoles with an arch and soft top cover on balance has been evaluated, and the result demonstrated that flat and soft insoles could be more advantageous for people which have balance problems with less possibility of decreasing postural balance [17]. Existing research investigated the effect of unilateral orthopaedic shoes with elevated soles and not offloading insoles, but they found significant asymmetry in joint range of motion which was not corrected by applying a twin shoe as a partner to orthopaedic shoes with an elevated sole [18]. Bruening et al. [19] evaluated the kinematics and mechanics of both conventional orthopaedic walking boots and a novel spring-loaded boot, and the findings revealed that all boots have an impact on the mechanics of the ankle joint in turn impacting on gait and balance. Whilst adapted footwear (not insoles) has been shown to change gait kinematics, the impact of insoles has mainly been neglected, despite the critical impact of altering balance, thus impacting mobility [17].

To the authors' knowledge, the effect of offloading insoles on gait kinematics, kinetics and plantar pressure has not been investigated. This study aims to investigate the effect of offloading insoles on plantar pressure and their influence on the gait kinematics. To the authors knowledge this is the first study which jointly measures both plantar pressure and gait kinematics using offloading insoles. The results should help to guide the future development and evaluation of offloading insoles that could reduce the risk of high plantar pressure and altered gait styles in a diabetic population. This is a pilot study and therefore only aims to measure the effects of the insole intervention rather than to solve any issues observed. The objectives of this study are: (a) evaluating the offloading insoles effect on high risk region; (b) estimating the offloading effect on gait kinematics and kinetics; (c) examining the subjects gait asymmetry change; (d) and analysing the relationship between the gait kinematics, kinetics and plantar pressure.

## Materials and methods

Ten participants were recruited for this experiment, and they were required to walk on a treadmill at three different speeds. During the experiment, six different offloading insoles were used to offload the metatarsal heads or calcaneus. The researchers measured various parameters, including gait kinematics, kinetics, ground reaction force, and plantar pressure. The analysis focused on average peak plantar pressure, pressure-time integral, lower limb sagittal angle, heel strike speed, toe-off speed, and centre of force.

## Participants and ethical approval

Ethical approval was granted for this experiment by the HRA and Health and Care Research Wales (HCRW) (application number 307041) and the trial registration identifier was NCT05865353 (ClinicalTrials.gov). Ten healthy participants (five male and five female) who did not have diabetes were recruited between 10[th] Sep 2022 and 1[st] October 2022 (sample size was chosen for convenience sampling similar to reference [20]). The exclusion criteria for the study are as follows: participants under the age of 18; individuals with any type of movement disorder; those who experience pain while walking; individuals with broken skin on their foot; and anyone suffering from severe skin conditions, including eczema, skin allergies or any kind of foot problems. All participants provided written informed consent which were stored in REDCap and could be accessed by contacting project Principal Investigator. The mean and standard deviation of age, height and weight of the participants was 25.4 (4.5) years old, 170 (6.5) cm and 68.3 (12.6) kg respectively. Subject foot sizes ranged from 5.5 to 11 (UK sizes).

## Footwear and offloading insoles

This experiment used diabetic shoes (97308, Finncomfort, Germany), characterized by a soft and shock-absorbing design, with a larger forefoot area than standard shoes and a flat midsole (Fig 1A). Participants were instructed to wear identical socks (SN: 0647207042, H&M Group, Sweden), made of 100% cotton and measuring approximately 1 mm in thickness. Insoles were custom-made, consisting of 1 mm layer of cellulose (Texon Cellulose Board, Algeo Ltd.), 5 mm layer of Aortha Medium Density EVA Foam (Aortha Medium Density EVA Foam, Algeo Ltd.), and 4 mm silicone layer (Smooth-Sil[tm] 950 Smooth-On, Inc.), from bottom to top (Fig 1B). Both the EVA layer and silicone layer were shore hardness A 50. The insoles were modified by removing circular sections of the middle EVA layer to create offloading regions (Fig 1B).

Six types of offloading insoles were manufactured including a control flat insole (Control), a small calcaneus offloading insole (SCO), a large calcaneus offloading insole (LCO), a small metatarsal head offloading insole (SMHO), a large metatarsal head offloading insole (LMHO), and a large calcaneus offloading insole for both feet (LCOBS) (Fig 1C). LCOBS was the symmetric condition with the LCO insoles in both left and right shoes. Although offloading insoles are normally prescribed asymmetrically, this study aims to understand the effect of offloading insoles on gait kinematics therefore this single symmetric condition was selected to study the effect of symmetrical insole conditions. The offloading region had a small size of 10 mm radius and a large size of 20 mm radius. All the conditions used a flat control insole on the left foot and an offloading insole on the right foot except LCOBS which had offloading insoles on both feet. The offloading apertures were located under 1[st] and 2[nd] metatarsal heads where DFU is mostly found [21]. Even though the occurrence of DFU at the calcaneus region is lower than metatarsal head region, the cost for calcaneus ulceration is 1.5 times that for metatarsal head ulceration and the limb salvage rate is 2–3 times less [22]. The poor healing rates and long healing period of calcaneus ulceration could be related to different mechanical loading and plantar skin healing potential [23]. Consequently, offloading apertures were also located at the calcaneus region.

## Offloading aperture location

Fig 1D shows the location of the calcaneus, 1st and 2nd metatarsal heads. The calcaneus location was 15% from the rear of the total foot length and central in the medial-lateral direction [24]. The 1st and 2nd metatarsal heads' location was determined using a pre-study on six healthy volunteers (four male and two female). The mean and standard deviation of age, height

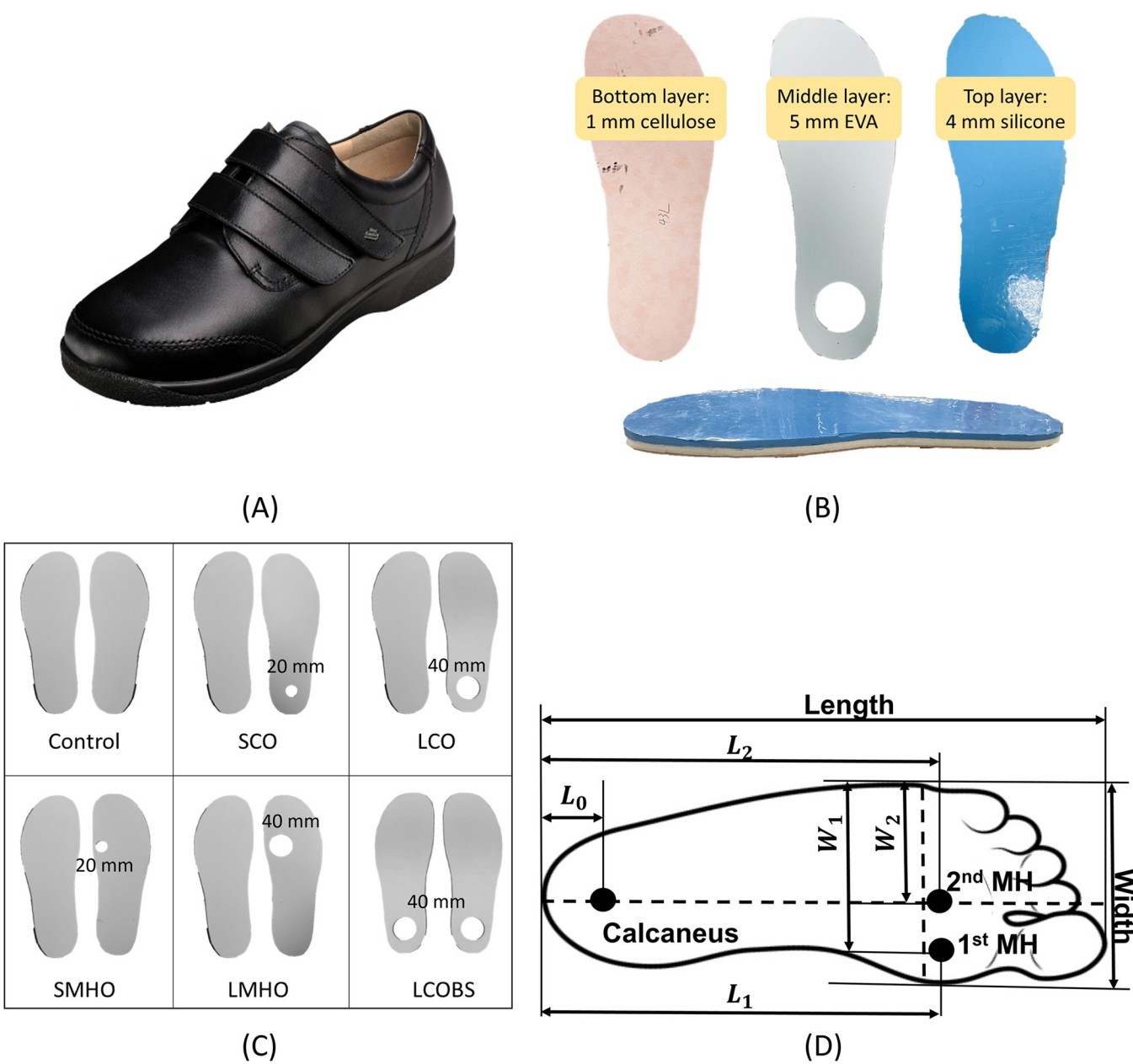

**Fig 1. Offloading insoles information.** (A) Diabetic shoes. (B) Offloading aperture locations. (C) The offloading insole configuration. The diameter of offloading apertures is given. (D) Measure of the offloading locations: L means the percentage of foot length and W means the percentage of the foot width.

and weight of the participants was 29.7 (7.6) years old, 179.8 (10.4) cm and 71 (20.8) kg respectively. The 1st and 2nd metatarsal heads' location was defined by manually palpating their feet to locate the bony anatomy, which was then marked with a pen. The subject then stood on a sheet of paper to transfer the ink marks to the paper to enable accurate measurements. The mean values of L1, W1, L2, and W2 were 71%, 85%, 74% and 63% respectively, where L is the percentage of foot length and W is the percentage of the foot width (Fig 1D). Both the small and large calcaneus offloading aperture centres were located at the calcaneus location. The small metatarsal head offloading aperture centres were located at 1st metatarsal head location, and the large metatarsal head offloading aperture centres were located at the mid-point

between 1st and 2nd metatarsal heads, which is 72.5% of the foot length and 74% of the foot width.

## Measurement of plantar pressure and gait kinematics during treadmill walking

Measurements of in shoe normal plantar pressure and gait kinematics were taken on different offloading insoles (Fig 1C). During the laboratory testing, the researcher utilized a premium wooden foot sizer & measuring device (JS Homewares, USA) to obtain measurements of the participants' foot size, subsequently selecting appropriate shoes and insoles for each participant. Participants were then instructed to wear diabetic shoes (97308, Finncomfort, Germany) and walk on the treadmill with the six different insole conditions (Fig 1C) for the experiment.

Prior to the commencement of walking on the split belt-instrumented treadmill with two force plates (1000 Hz, M-Gait, Motek Medical BV, Amsterdam, Netherlands), reflective markers were affixed (Fig 2). A total of 16 markers were affixed to the diabetic shoes, excluding the body markers (Fig 2). The locations of the markers were recorded utilizing a 12-camera motion capture system (100 Hz, Miqus M3, Qualisys AB, Gothenburg, Sweden). In addition, an in-shoe pressure measurement system (100Hz, F-Scan, Tekscan Inc., Norwood, MA, USA) was inserted into the diabetic shoes and located above the insoles. The treadmill, motion capture system, and plantar pressure measurement system were facilitated by D-Flow (v3.34.0), Qualisys Track Manager (v2021.2), and F-Scan research (v7.00–19), respectively. The synchronization between the two force plates and the motion capture system was achieved through a hardwired connection. The initial heel strike was utilised as a common event to synchronize the in-shoe pressure measurement system with both the force plates and the motion capture system.

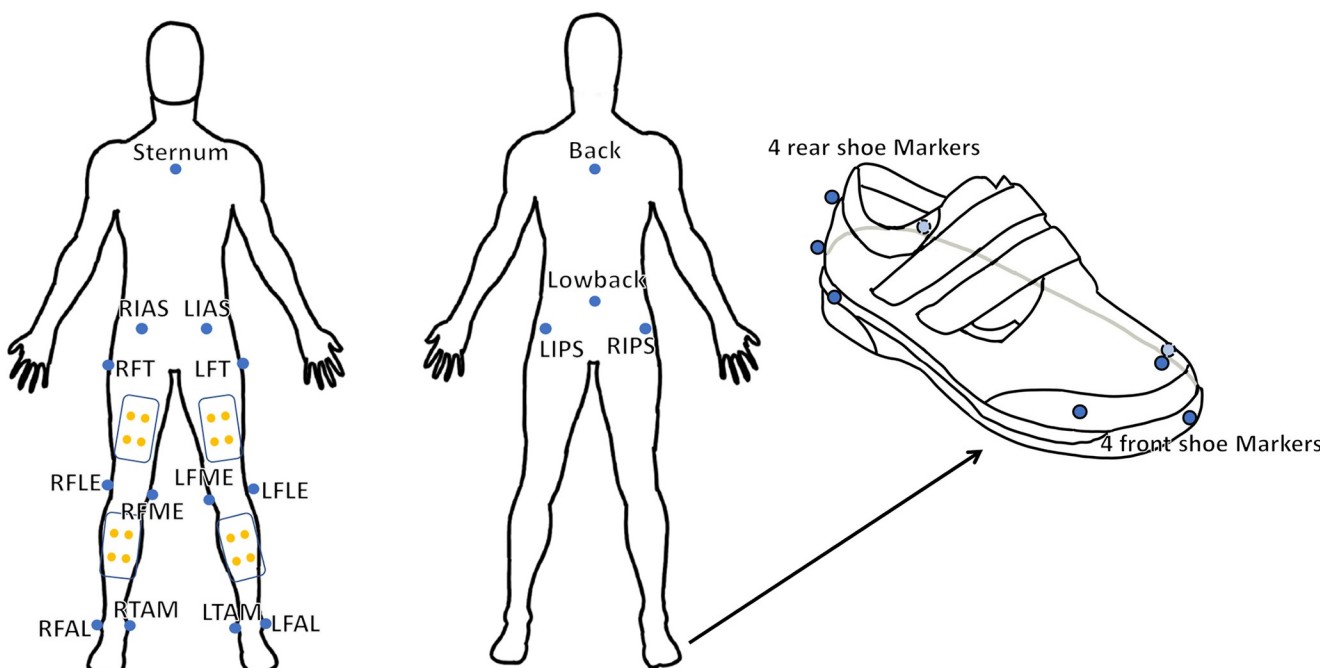

**Fig 2. Reflective markers.** Reflective markers are attached to the subjects' bodies. The markers are located at ilium anterior superior, ilium posterior superior, femur greater trochanter, femur lateral epicondyle, femur medial epicondyle, fibula apex of lateral malleolus, tibia apex of medial malleolus, sternum jugular notch, back and low back. Eight markers are located at each diabetic shoe.

Participants had a 5-minute acclimatisation walking period to the new offloading insoles before capturing data. The participants were required to walk on the treadmill to determine their normal walking speed (NWS) which was their self-selected walking speed to reflect their preferred walking pace. Subjects then walked on the treadmill at a slow speed (NWS—0.2 m/s), then normal speed (NWS) and finally fast speed (NWS + 0.2 m/s) for 3 minutes and 20 seconds each (total duration 10 minutes). The participants started from rest on the treadmill and the speed was changed through normal, fast and slow walking speeds. During this time, data was collected from the optical cameras of the motion capture system through skin-mounted reflective markers, pressure-sensing insoles, and force plates in the split-belt treadmill. The insoles were then changed by the researcher for one of the six insoles and the conditions were recorded by the researcher. The test was repeated until the participant has been tested for all six insole conditions. The researcher applied a MATLAB code to generate the random allocation sequence of offloading insole conditions. Participants were not provided with information about the conditions of offloading insoles during the whole experiment, and the offloading insole conditions cannot be visually distinguished because the offloading layer is covered by flat bottom and top layers.

## Data analysis

The data from the 10 participants were evaluated on an individual basis using the data from the individuals control insole condition as a baseline to compare kinematic and pressure differences. The region of interest (RoI) and the region of the foot (RoF) is defined (Fig 3). The F-Scan research software (v7.00–19) was utilized to post-process the plantar pressure data. The data was inspected and no variation in the plantar pressure data was found, so representative 10 cycles were taken for analysis. Data of 10 gait cycles starting from the first gait cycle after 20 seconds of normal speed walking for most of the subjects were exported. Average peak plantar pressure (APPP) in the calcaneus, metatarsal head, and toe regions was exported. For each subject, the percentage difference in APPP between the control insole and five offloading insoles at RoI and RoF was calculated and then the mean data of 10 subjects were calculated.

The pressure time integrals (PTI) of 10 gait cycles starting from the first gait cycle after 20 seconds of normal speed walking for most of the subjects in the calcaneus, metatarsal head, and toe region was exported from F-Scan research software and was calculated using MATLAB R2021a. For each subject, the percentage difference in PTI between the control insole and five offloading insoles at RoI and RoF were calculated and then the mean data of 10 subjects were calculated.

The arch index was calculated from normal plantar pressure measurements taken with the F-scan system, using the method outlined in reference [24]. In addition, the total contact area of the foot was recorded and exported.

The kinematic data were exported from the Qualisys Track Manager software (v2021.2) and post-processed using Visual3D (x64 Professional v2022.08.3). A low limb model was created using Visual3D, and the 'Automatic Gait Events' function was applied to define heel strike and toe-off events, with manual review and improvement to ensure accuracy. The heel strike and toe-off were defined as the time points when the ground reaction force increases from zero and decreases to zero respectively. The data was inspected and no variation in the gait data was found, so representative 10 cycles were taken for analysis. The beginning of this period was after the subject had been walking for approximately 20 seconds at constant speed. The kinematic data was then exported and processed using MATLAB R2021a. The right thigh angle, right shank angle, right foot angle, right knee angle, right ankle angle, and right shank distal end velocity during both heel strike and toe-off were reviewed. The right thigh, shank

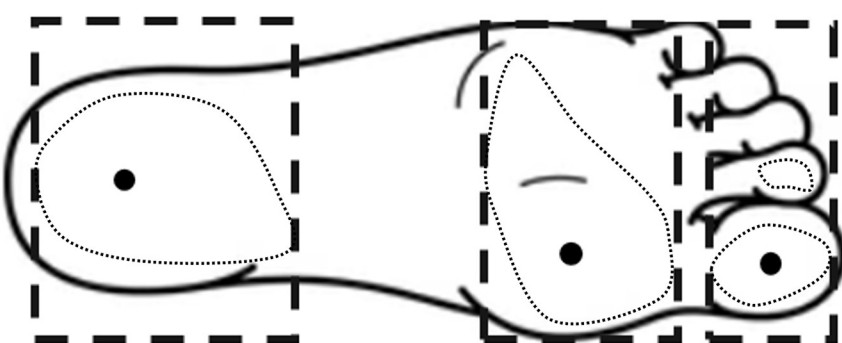

● Max APPP location at calcaneus, metatarsal heads and toes, where is define as region of interest (ROI)

⬭ Regions where the APPP is larger than 100 kPa

⸥⸤ Region of foot (ROF) including calcaneus, metatarsal head and toe regions which are defined by the regions where the APPP is larger than 100 kPa

**Fig 3. The definition of region of interest (RoI) and region of foot (RoF).** The three black dots are the location of maximum average peak plantar pressure (APPP) of control insoles, which were defined as RoI for all offloading conditions of each subject. The dotted lines are the regions where the APPP is larger than 100 kPa. The dashed line rectangles are calcaneus, metatarsal head and toe regions which are defined by the regions where the APPP is larger than 100 kPa, and they were defined as RoF (calcaneus region, metatarsal head region and toe region). The RoI and RoF when analysing pressure time integral has the same definition.

and foot angles are the angles between the segments and global lab coordinates frames around the x-axis (perpendicular to travel direction on the ground). The right knee angle is the angle between the right thigh and shank around the x-axis, and the right ankle angle is the angle between the right shank and foot around the x-axis. For each subject, the difference in angle between the control insole and five offloading insoles were calculated and then the mean data of all 10 subjects was also calculated.

The centre of force (CoF) was calculated from the treadmill force plates (1000 Hz, M-Gait, Motek Medical BV, Amsterdam, Netherlands) using the Eq 1 and the centre of the CoF was defined as the average value in both the x and y axis.

$$CoF = \frac{LoC_l \times GRF_l + LoC_r \times GRF_r}{GRF_l + GRF_r} \tag{1}$$

where $LoC$ means the location of force referring to the centre of treadmill, $GRF$ means the magnitude of normal ground reaction force on the treadmill, subscript $l$ means left, and subscript $r$ means right. The width and length of the CoF were analysed by MATLAB R2021a (Fig 4).

Walking asymmetry of width and length was calculated using Eq 2 and 3.

$$Asymmetry\ of\ width\ (\%) = \frac{100 \times |W_l - W_r|}{0.5 \times |W_l + W_r|} \tag{2}$$

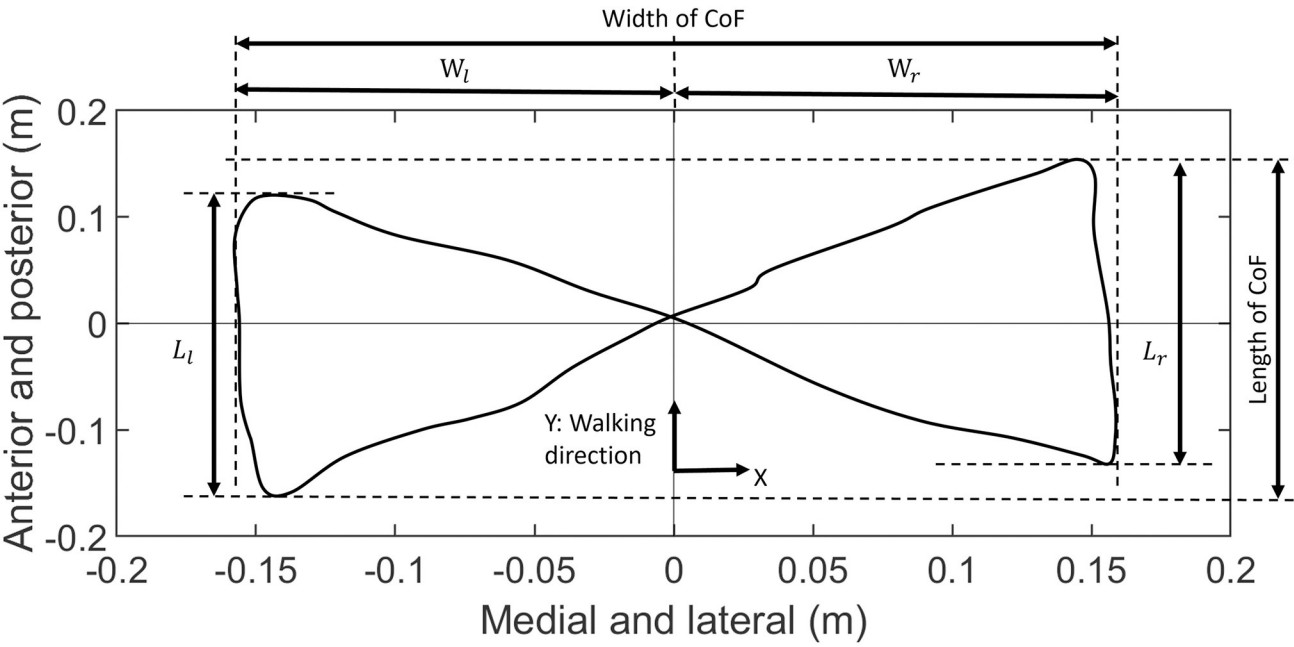

**Fig 4. Width and length of centre of force (CoF) calculated from the two split treadmill force plates data.** $W_l$ is left width of CoF, $W_r$ is right width of CoF, $L_l$ is left length of CoF, and $L_r$ is right length of CoF. Y-axis is the walking direction (the positive direction is anterior), X-axis is the medial and lateral direction (the positive direction is right hand side), and Z-axis vertical to X-axis and Y-axis is the standing direction.

$$Asymmetry\ of\ length\ (\%) = \frac{100 \times |L_l - L_r|}{0.5 \times |L_l + L_r|} \tag{3}$$

Where W means CoF width, L means CoF length, subscript l means left, and subscript r means right. For each subject, the percentage difference in CoF width and length between the control insole and five offloading insoles were calculated and then the mean data of 10 subjects were calculated, and the difference in asymmetry of width and length was calculated.

## Statistical methods

Statistical analyses were conducted using IBM SPSS Statistics (version 29.0.2.0, IBM SPSS, Inc., Chicago, IL, USA). Prior to analysis, A Shapiro-Wilk test was conducted to assess the normality of the data. The results of the Shapiro-Wilk test confirmed that the data were normally distributed. A paired t-test was then conducted to compare the plantar pressure and kinematic data between five offloading and one control insole conditions.

## Results

### Average peak plantar pressure and pressure time integral

Fig 5 presents the average peak plantar pressure (APPP) of one subject under five offloading conditions and a control condition, each recorded at a normal walking speed. Table 1 shows the mean average peak plantar pressure at region of foot (RoF) and region of interest (RoI) (defined in Fig 3) and shank distal end velocity of five offloading conditions and the control condition for 10 subjects at three walking speeds. Percentage differences and P-values were calculated for the five offloading conditions and the control condition for 10 subjects at three

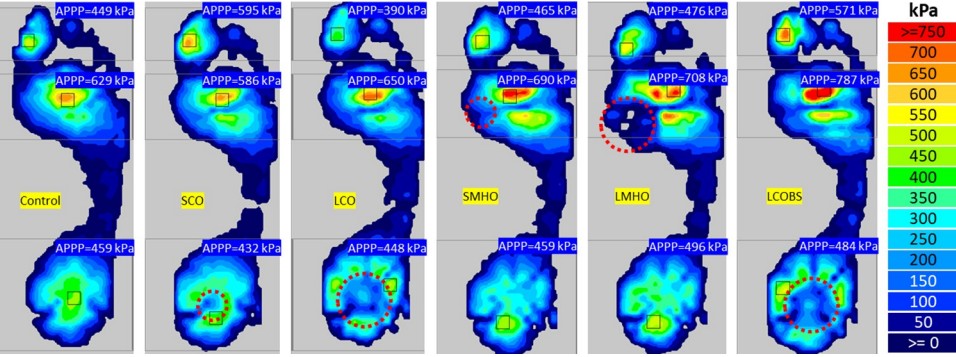

**Fig 5. The average peak plantar pressure of one subject under five offloading conditions and a control condition, each recorded at a normal walking speed.** The red dash cycle is the location of offloading aperture.

walking speeds (Table 1). Only LCO and LCOBS showed a significant decrease in APPP in the region of interest at calcaneus when walking at all three speeds (P<0.05). Both LCO and SCO insoles could decrease the APPP at the RoI. When walking at slow, normal, and fast speed, LCO insoles could decrease more calcaneus APPP by a mean of -62% (range -57% to -68%, P<0.05) at the RoI than small calcaneus region offloading (SCO) insoles by a mean of -18% (-14% to -23%, P = 0.040, 0.108, and 0.264), but LCO insoles caused larger APPP increase of a mean of 23% (21% to 26%, P<0.05) in metatarsal head region at three walking speed. Calcaneus offloading insoles increased the APPP in calcaneus region (outside of the RoI) by a mean of 10% (range -1% to 17%) except large calcaneus offloading (LCO) insoles during slow walking while only SCO insoles at normal and fast walking speed observed significant increases (P = 0.002 and 0.001). Calcaneus offloading insoles increased the APPP in metatarsal head region by a mean of 15% (range 4% to 26%) while only LCO insoles at three walking speed observed significant increases (P<0.05). Both small metatarsal head offloading (SMHO) and large metatarsal head offloading (LMHO) insoles could decrease the APPP at the metatarsal head RoI by a mean of -11% (range -3% to -21%, P>0.05) and LMHO decreased more during slow and fast walking. Metatarsal head offloading insoles increased the APPP in calcaneus region by a mean of 15% (range 11% to 19%) while the increase was only significant at fast walking speed (P<0.05), and they increased the APPP in metatarsal head region by a mean of 18% (range 8% to 34%, all P>0.05 except LMHO at normal walking speed). Large both-side calcaneus offloading (LCOBS) and LCO insoles had similar offloading effects at calcaneus RoI. LCOBS insoles observed larger APPP increase in metatarsal head region during normal and fast walking. All offloading insoles increased the APPP in toe region during slow and fast walking.

Fig 6 presents the plantar pressure and the pressure time integral in the regions of interest at the calcaneus and metatarsal heads during one gait cycle of ten subjects walking at normal speed. Fig 6 shows that the SCO, LCO, and LCOBS insoles decreased the peak plantar pressure in calcaneus region, while no difference of metatarsal heads peak plantar pressure was observed when utilising the SMHO and LMHO insoles. Additionally, all the conditions show differences in peak plantar pressure profiles across the gait cycle for either the calcaneus or metatarsal head region. Table 2 shows the mean pressure time integral (PTI) at region of foot (RoF) and region of interest (RoI) (defined in Fig 3) and shank distal end velocity of five offloading conditions and the control condition for 10 subjects at three walking speeds. Percentage differences and P-values were calculated for the five offloading conditions and the control condition for 10 subjects at three walking speeds (Table 2). LCO insoles could decrease more

**Table 1. The mean average peak plantar pressure and shank distal end velocity of five offloading conditions and the control condition at the region of foot (RoF) and region of interest (RoI) for 10 subjects at three walking speeds.**

| Walking speed | Offloading condition | Velocity (m/s, mean[std]) | | APPP at region of foot (kPa, mean[std]) | | | APPP at region of interest (kPa, mean[std]) | | |
|---|---|---|---|---|---|---|---|---|---|
| | | Heel strike | Toe-off | Calcaneus | Metatarsal head | Toe | Calcaneus | Metatarsal head | Toe |
| Slow | Control | 0.60[0.08] | 0.96[0.09] | 469[107] | 530[166] | 512[236] | 469[107] | 530[166] | 512[236] |
| | SCO | 0.67[0.10] | 1.06[0.08] | 495[115] | 570[241] | 482[206] | 367[164] | 490[235] | 380[254] |
| | LCO | 0.79[0.15] | 1.11[0.06] | 463[157] | 672[340] | 499[241] | 164[104] | 586[403] | 367[291] |
| | SMHO | 0.57[0.06] | 0.95[0.12] | 512[115] | 562[151] | 535[218] | 405[97] | 437[189] | 346[249] |
| | LMHO | 0.68[0.08] | 1.09[0.12] | 522[121] | 573[152] | 546[221] | 428[88] | 398[231] | 352[245] |
| | LCOBS | 0.79[0.14] | 1.10[0.08] | 426[110] | 615[223] | 466[206] | 176[112] | 556[258] | 331[239] |
| Normal | Control | 0.61[0.11] | 0.96[0.12] | 490[117] | 535[165] | 681[331] | 490[117] | 535[165] | 681[331] |
| | SCO | 0.69[0.10] | 1.09[0.07] | 551[134] | 559[175] | 613[209] | 405[196] | 492[110] | 432[219] |
| | LCO | 0.79[0.12] | 1.11[0.09] | 566[178] | 654[201] | 585[247] | 146[44] | 602[180] | 413[288] |
| | SMHO | 0.60[0.08] | 0.98[0.08] | 553[143] | 579[146] | 611[254] | 474[128] | 469[199] | 385[292] |
| | LMHO | 0.70[0.16] | 1.05[0.06] | 557[154] | 722[289] | 691[271] | 454[129] | 484[260] | 418[266] |
| | LCOBS | 0.74[0.11] | 1.09[0.10] | 541[174] | 724[299] | 627[240] | 149[65] | 652[326] | 350[322] |
| Fast | Control | 0.65[0.11] | 0.97[0.10] | 522[113] | 596[228] | 671[253] | 522[113] | 596[228] | 671[253] |
| | SCO | 0.73[0.17] | 1.12[0.10] | 608[131] | 639[219] | 718[292] | 449[207] | 571[170] | 477[299] |
| | LCO | 0.75[0.13] | 1.10[0.06] | 571[215] | 646[195] | 687[285] | 213[75] | 584[209] | 493[350] |
| | SMHO | 0.62[0.11] | 0.98[0.13] | 603[145] | 652[155] | 693[256] | 535[156] | 519[179] | 477[294] |
| | LMHO | 0.66[0.09] | 1.06[0.05] | 616[127] | 745[265] | 715[225] | 534[123] | 453[300] | 512[308] |
| | LCOBS | 0.80[0.11] | 1.14[0.07] | 501[99] | 777[330] | 676[242] | 222[94] | 715[357] | 320[187] |
| Walking speed | Offloading condition | Velocity (Control value, m/s; difference from control, % and [P-value]) | | APPP at region of foot (Control value, kPa; difference from control, % and [P-value]) | | | APPP at region of interest (Control value, kPa; difference from control, % and [P-value]) | | |
| | | Heel strike | Toe-off | Calcaneus | Metatarsal head | Toe | Calcaneus | Metatarsal head | Toe |
| Slow | Control | 0.6 | 0.96 | 469 | 530 | 512 | 469 | 530 | 512 |
| | SCO | 12[0.091] | 11[0.000] | 6[0.175] | 4[0.234] | 9[0.673] | -23[0.040] | -11[0.426] | -14[0.148] |
| | LCO | 30[0.003] | 17[0.001] | -1[0.892] | 21[0.049] | 14[0.752] | -62[0.001] | -1[0.521] | -14[0.188] |
| | SMHO | -6[0.117] | -2[0.598] | 11[0.189] | 8[0.265] | 24[0.730] | -13[0.006] | -12[0.294] | -8[0.157] |
| | LMHO | 15[0.020] | 15[0.044] | 13[0.096] | 10[0.240] | 24[0.665] | -7[0.121] | -21[0.195] | -10[0.170] |
| | LCOBS | 29[0.004] | 17[0.010] | -7[0.277] | 16[0.031] | 6[0.535] | -59[0.001] | 2[0.581] | -18[0.104] |
| Normal | Control | 0.61 | 0.96 | 490 | 535 | 681 | 490 | 535 | 681 |
| | SCO | 15[0.005] | 16[0.001] | 13[0.002] | 6[0.192] | -2[0.327] | -19[0.108] | -5[0.104] | -31[0.030] |
| | LCO | 32[0.002] | 17[0.004] | 17[0.142] | 26[0.014] | -2[0.379] | -68[0.001] | 16[0.039] | -29[0.065] |
| | SMHO | 1[0.909] | 3[0.398] | 15[0.149] | 12[0.158] | 0[0.256] | -3[0.580] | -7[0.429] | -32[0.067] |
| | LMHO | 15[0.004] | 11[0.006] | 14[0.091] | 34[0.028] | 21[0.809] | -8[0.192] | -7[0.579] | -18[0.113] |
| | LCOBS | 25[0.008] | 15[0.013] | 14[0.423] | 36[0.015] | 10[0.668] | -67[0.001] | 18[0.110] | -33[0.073] |
| Fast | Control | 0.65 | 0.97 | 522 | 596 | 671 | 522 | 596 | 671 |
| | SCO | 12[0.006] | 16[0.004] | 17[0.001] | 9[0.146] | 9[0.366] | -14[0.264] | 0[0.674] | -25[0.085] |
| | LCO | 23[0.026] | 16[0.005] | 7[0.485] | 22[0.025] | 3[0.851] | -57[0.001] | 9[0.319] | -30[0.083] |
| | SMHO | -1[0.755] | 2[0.426] | 17[0.038] | 14[0.240] | 9[0.962] | 2[0.666] | -3[0.488] | -22[0.093] |
| | LMHO | 1[0.945] | 11[0.023] | 19[0.004] | 30[0.164] | 20[0.673] | 3[0.552] | -16[0.291] | -10[0.270] |
| | LCOBS | 24[0.001] | 19[0.002] | 0[0.661] | 30[0.002] | 13[0.967] | -55[0.001] | 18[0.072] | -40[0.020] |

Percentage differences and P-values were calculated for the five offloading conditions and the control condition for 10 subjects at three walking speeds. P-values smaller than 0.05 were underlined. APPP means average peak plantar pressure, SCO means small calcaneus offloading insole, LCO means larger calcaneus offloading insole, SMHO means small metatarsal head offloading insole, LMHO means large metatarsal head offloading insole, LCOBS means both sides large calcaneus offloading insole and std means standard deviation.

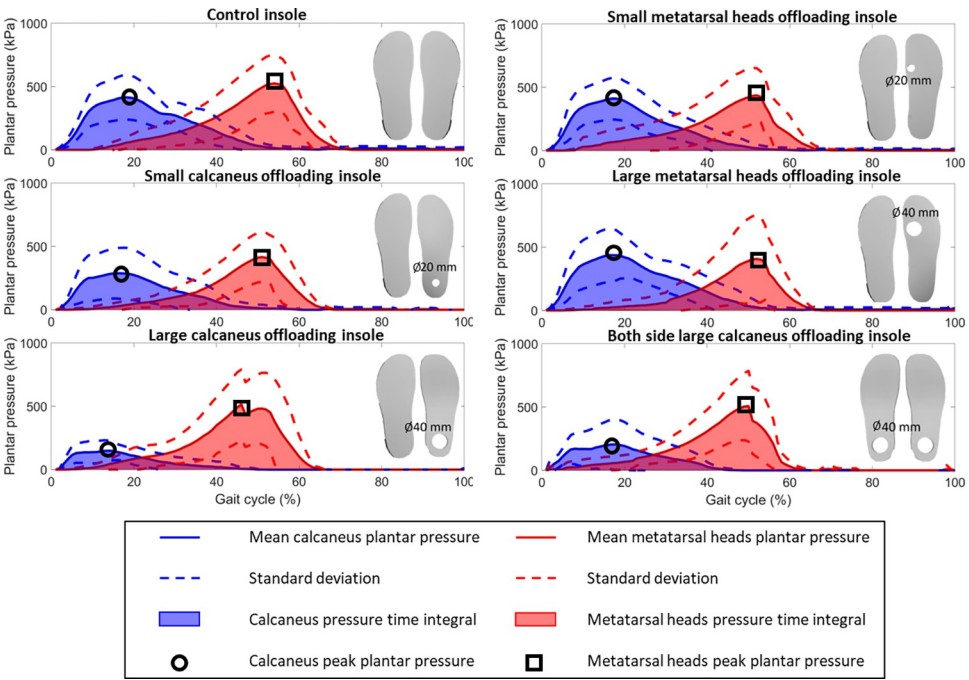

**Fig 6. The plantar pressure and the pressure time integral in the regions of interest at the calcaneus and metatarsal heads during one gait cycle of ten subjects walking at normal speed.** The black circles and squares indicate the locations of peak plantar pressure at the calcaneus and metatarsal heads regions of interest respectively, while the blue and red shadows represent the pressure time integral at the calcaneus and metatarsal heads regions of interest, respectively.

calcaneus PTI by a mean of -70% (range -66% to -75%, P = 0.000, 0.000, and 0.389) at the calcaneus RoI than SCO insoles by a mean of -20% (range -13% to -30%, P = 0.025, 0.321, and 0.185), but they caused larger PTI increase in metatarsal head region during slow and normal walking. SCO insoles increased the PTI in calcaneus region by a mean of 10% (range 4% to 14%) while only observed the significant increase at normal walking speed (P = 0.039). Calcaneus offloading insoles observed the PTI increase in metatarsal head region except SCO during normal walking speed (P>0.05). Metatarsal head offloading insoles decreased the PTI at the metatarsal head RoI by a mean of -23% (range -7% to -32%, P>0.05) but increased the PTI in metatarsal head region by a mean of 7% (range 3% to 12%, P>0.05). LMHO insoles observed a larger decrease of PTI at metatarsal head RoI than SMHO.

### Gait kinematics

All offloading insoles except SMHO increased the heel strike velocity by a mean of 19% (range 1% to 32%, all P <0.05 except SCO at slow walking speed, SMHO at all walking speed, and LMHO at fast walking speed), as shown in Table 1. Toe-off velocity was observed to increase significantly by a mean of 15% (range 11% to 19%, P<0.05) when applying offloading insoles except SMHO. LCO increased the heel strike velocity and toe off velocity by a mean of 23% (range 16% to 32%, P<0.05) during walking at three speeds. LMHO also observed increases in heel strike velocity and toe off velocity during slow and normal walking (P<0.05). No difference in the right thigh, shank and foot angle was observed. Metatarsal head offloading insoles observed larger right thigh heel strike angles than calcaneus offloading insoles. LCOBS observed larger right thigh heel strike angle change than LCO, and the largest change (1.47°)

**Table 2. The mean pressure time integrals and shank distal end velocity of five offloading conditions and the control condition at the region of foot (RoF) and region of interest (RoI) for 10 subjects at three walking speeds.**

| Walking speed | Offloading condition | Velocity (m/s, mean[std]) | | PTI at region of foot (kPa·s, mean[std]) | | | PTI at region of interest (kPa·s, mean[std]) | | |
|---|---|---|---|---|---|---|---|---|---|
| | | Heel strike | Toe-off | Calcaneus | Metatarsal head | Toe | Calcaneus | Metatarsal head | Toe |
| Slow | Control | 0.60[0.08] | 0.96[0.09] | 1726[293] | 1587[538] | 1009[668] | 1726[293] | 1587[538] | 1009[668] |
| | SCO | 0.67[0.10] | 1.06[0.08] | 1781[340] | 1705[821] | 917[497] | 1171[533] | 1395[599] | 656[524] |
| | LCO | 0.79[0.15] | 1.11[0.06] | 1628[534] | 1885[1121] | 897[495] | 406[179] | 1627[1323] | 528[475] |
| | SMHO | 0.57[0.06] | 0.95[0.12] | 1938[310] | 1579[434] | 978[549] | 1548[306] | 1109[469] | 425[427] |
| | LMHO | 0.68[0.08] | 1.09[0.12] | 1915[348] | 1542[480] | 1007[633] | 1622[266] | 1024[733] | 533[486] |
| | LCOBS | 0.79[0.14] | 1.10[0.08] | 1547[400] | 1757[791] | 843[586] | 506[298] | 1483[797] | 428[316] |
| Normal | Control | 0.61[0.11] | 0.96[0.12] | 1457[197] | 1489[596] | 1402[982] | 1457[197] | 1489[596] | 1402[982] |
| | SCO | 0.69[0.10] | 1.09[0.07] | 1648[359] | 1488[667] | 1090[618] | 1285[563] | 1182[368] | 574[326] |
| | LCO | 0.79[0.12] | 1.11[0.09] | 1578[433] | 1553[448] | 996[493] | 475[285] | 1233[657] | 527[408] |
| | SMHO | 0.60[0.08] | 0.98[0.08] | 1659[289] | 1501[524] | 883[298] | 1197[471] | 1100[737] | 440[336] |
| | LMHO | 0.70[0.16] | 1.05[0.06] | 1574[368] | 1508[534] | 1071[507] | 1145[442] | 983[683] | 580[510] |
| | LCOBS | 0.74[0.11] | 1.09[0.10] | 1513[501] | 1850[960] | 866[319] | 371[301] | 1585[954] | 430[341] |
| Fast | Control | 0.65[0.11] | 0.97[0.10] | 1352[217] | 1443[637] | 1094[501] | 1352[217] | 1443[637] | 1094[501] |
| | SCO | 0.73[0.17] | 1.12[0.10] | 1530[305] | 1537[665] | 982[346] | 1147[476] | 1259[478] | 592[499] |
| | LCO | 0.75[0.13] | 1.10[0.06] | 1356[435] | 1335[462] | 903[462] | 412[240] | 1127[439] | 622[416] |
| | SMHO | 0.62[0.11] | 0.98[0.13] | 1530[243] | 1417[384] | 971[330] | 1210[235] | 1131[456] | 542[287] |
| | LMHO | 0.66[0.09] | 1.06[0.05] | 1624[192] | 1449[315] | 1045[399] | 1306[172] | 934[699] | 687[462] |
| | LCOBS | 0.80[0.11] | 1.14[0.07] | 1116[226] | 1691[723] | 978[471] | 475[326] | 1418[808] | 430[347] |
| Walking speed | Offloading condition | Velocity (Control value, m/s; *difference from control, % and [P-value]*) | | PTI at region of foot (Control value, kPa·s; *difference from control, % and [P-value]*) | | | PTI at region of interest (Control value, kPa·s; *difference from control, % and [P-value]*) | | |
| | | Heel strike | Toe-off | Calcaneus | Metatarsal head | Toe | Calcaneus | Metatarsal head | Toe |
| Slow | Control | 0.6 | 0.96 | 1726 | 1587 | 1009 | 1726 | 1587 | 1009 |
| | SCO | 12[0.091] | 11[0.000] | 4[0.581] | 4[0.679] | 15[0.480] | -29[0.025] | -12[0.345] | -23[0.442] |
| | LCO | 30[0.003] | 17[0.001] | -6[0.437] | 15[0.489] | 16[0.314] | -75[0.000] | -4[0.347] | -29[0.532] |
| | SMHO | -6[0.117] | -2[0.598] | 14[0.061] | 3[0.899] | 27[0.354] | -9[0.082] | -20[0.218] | -33[0.394] |
| | LMHO | 15[0.020] | 15[0.044] | 12[0.104] | 3[0.952] | 26[0.338] | -3[0.379] | -26[0.259] | -23[0.743] |
| | LCOBS | 29[0.004] | 17[0.010] | -9[0.151] | 15[0.616] | 0[0.867] | -68[0.000] | -4[0.471] | -31[0.291] |
| Normal | Control | 0.61 | 0.96 | 1457 | 1489 | 1402 | 1457 | 1489 | 1402 |
| | SCO | 15[0.005] | 16[0.001] | 12[0.039] | -0[0.993] | -7[0.487] | -12[0.321] | -15[0.046] | -45[0.040] |
| | LCO | 32[0.002] | 17[0.004] | 7[0.301] | 11[0.540] | -5[0.682] | -65[0.000] | -17[0.111] | -41[0.074] |
| | SMHO | 1[0.909] | 3[0.398] | 14[0.055] | 8[0.950] | -12[0.460] | -19[0.054] | -18[0.232] | -50[0.039] |
| | LMHO | 15[0.004] | 11[0.006] | 8[0.278] | 6[0.916] | 12[0.970] | -22[0.032] | -32[0.091] | -29[0.141] |
| | LCOBS | 25[0.008] | 15[0.013] | 4[0.731] | 28[0.158] | -11[0.469] | -73[0.000] | 4[0.634] | -49[0.042] |
| Fast | Control | 0.65 | 0.97 | 1352 | 1443 | 1094 | 1352 | 1443 | 1094 |
| | SCO | 12[0.006] | 16[0.004] | 14[0.051] | 7[NA] | 1[0.357] | -16[0.185] | -8[0.970] | -42[0.017] |
| | LCO | 23[0.026] | 16[0.005] | -1[0.925] | 6[NA] | -11[0.222] | -69[0.389] | -10[0.742] | -33[0.369] |
| | SMHO | -1[0.755] | 2[0.426] | 15[0.063] | 8[NA] | 8[0.506] | -9[0.081] | -7[0.113] | -34[0.033] |
| | LMHO | 1[0.945] | 11[0.023] | 22[0.002] | 12[NA] | 20[0.825] | -1[0.537] | -27[0.738] | -16[0.138] |
| | LCOBS | 24[0.001] | 19[0.002] | -13[0.078] | 24[NA] | 8[0.598] | -65[0.000] | -1[0.864] | -45[0.017] |

Percentage differences and P-values were calculated for the five offloading conditions and the control condition for 10 subjects at three walking speeds. P-values smaller than 0.05 were underlined. PTI means pressure time integral, SCO means small calcaneus offloading insole, LCO means larger calcaneus offloading insole, SMHO means small metatarsal head offloading insole, LMHO means large metatarsal head offloading insole, LCOBS means both sides large calcaneus offloading insole, and std means standard deviation.

was found at LCOBS of normal speed walking. Metatarsal head offloading insoles observed larger right shank heel strike angle than calcaneus offloading insoles.

Table 3 shows the mean value of width of centre of force (CoF), length of CoF, and asymmetry of width and length of five offloading conditions and control condition of 10 subjects at three walking speeds. Percentage differences and P-values were calculated for the five offloading conditions and the control condition for 10 subjects at three walking speeds (Table 3). No clear trend was observed in the width of CoF but all offloading insoles increased the length of CoF by a mean of 4% (range 1% to 7%). All offloading insoles induced an effect on the asymmetry of width, with the presence of larger offloading apertures leading to a more asymmetry gait.

Table 4 shows the mean arch index and total contact area. P-values were calculated for the five offloading conditions and the control condition for the 10 subjects at three walking speeds (Table 4). There was a statistical trend that showed decreases in the arch index for the SCO condition during fast walking (-7%, P = 0.034), the SMHO condition during fast walking (-9%, P = 0.035), and the LMHO condition during slow and normal walking (-9%, P = 0.011; -14%, P = 0.035). A statistical trend showed decreases in the total contact area for the SCO condition during slow walking (-4%, P = 0.001) and the LMHO condition during slow and normal walking (-5%, P = 0.031; -6%, P = 0.049).

### Relationship between plantar pressure and kinematics

When walking at slow and normal speeds, LCO observed a larger toe-off velocity than SCO, and the larger toe-off velocity corresponded to a larger APPP in metatarsal head region, see Table 1. However, it is important to note that this trend was not observed during fast walking. LMHO observed a larger heel strike speed than SMHO and the larger heel strike velocity indicated the larger APPP in calcaneus region. This trend is not observed when walking at normal and fast speeds.

### Discussion

The aim of this study was to investigate the effect of offloading insoles on plantar pressure and their influence on the gait kinematics. To the authors knowledge, it is the first time to jointly measure gait kinematics and plantar pressure changes caused by offloading insoles which off-load high-risk regions by removing insole material. The results demonstrate that the offloading insoles can decrease the high-risk region plantar pressure, which correlates with other studies [9,10], if the offloading aperture centre is collocated with the subjects' peak high plantar pressure point. But the offloading insoles reveal a negative effect on the kinematics including an increase in heel strike and toe-off velocity, which could cause plantar pressure increases, and an increase of walking asymmetry of width. The increased heel strike velocity could reduce pressure time integrals. However, the increased foot loading velocity could prove a challenge to control the foot flat for the dorsiflexors, and in patients with diabetes which might result or increase the risk of a 'foot slap' landing.

The offloading insoles can decrease both the average peak plantar pressure and pressure time integrals at the region of interest. Lin et al. [9] and Penny et al. [10] reported plantar pressure decrease when applying offloading insoles but they did not review the pressure time integrals. However, after applying the offloading aperture, the plantar pressure around offloading aperture edge is found to increase, which might cause a new high risk DFU area. Armstrong et al. [13] predicted edge effect is involved with shear and normal stress. Penny et al. [10] also observed the edge effect when applying offloading insoles. The edge effect caused by the offloading insoles indicate the importance to optimise the shape of offloading apertures. Shaulian

**Table 3. The mean centre of force width and length, and asymmetry of width and length of five offloading conditions and the control condition for 10 subjects at three walking speeds.**

| | | Mean and standard deviation data | | | |
|---|---|---|---|---|---|
| Walking speed | Offloading condition | Width (m, mean[std]) | Length (m, mean[std]) | Asymmetry of width (mean [std]) | Asymmetry of length (mean [std]) |
| Slow | Control | 0.21[0.04] | 0.31[0.05] | 5.90[4.52] | 11.33[4.70] |
| | SCO | 0.21[0.03] | 0.32[0.04] | 4.60[0.68] | 13.70[7.27] |
| | LCO | 0.22[0.04] | 0.33[0.05] | 5.41[2.11] | 12.29[4.16] |
| | SMHO | 0.21[0.04] | 0.33[0.05] | 4.75[2.10] | 11.07[4.23] |
| | LMHO | 0.21[0.04] | 0.33[0.06] | 4.85[1.58] | 13.06[5.05] |
| | LCOBS | 0.21[0.04] | 0.33[0.05] | 5.42[2.09] | 12.69[2.74] |
| Normal | Control | 0.22[0.04] | 0.35[0.06] | 4.81[1.26] | 9.15[2.60] |
| | SCO | 0.22[0.04] | 0.36[0.06] | 4.84[1.66] | 9.10[3.55] |
| | LCO | 0.22[0.04] | 0.35[0.06] | 5.07[2.08] | 7.38[2.23] |
| | SMHO | 0.22[0.03] | 0.37[0.05] | 5.70[2.10] | 9.03[3.21] |
| | LMHO | 0.22[0.03] | 0.37[0.06] | 6.35[2.34] | 8.90[4.65] |
| | LCOBS | 0.21[0.04] | 0.36[0.06] | 4.63[1.27] | 8.90[3.14] |
| Fast | Control | 0.21[0.03] | 0.41[0.06] | 5.08[1.65] | 7.18[1.91] |
| | SCO | 0.21[0.02] | 0.41[0.06] | 5.78[1.84] | 8.69[2.44] |
| | LCO | 0.22[0.03] | 0.42[0.06] | 6.96[4.32] | 8.33[4.61] |
| | SMHO | 0.21[0.03] | 0.42[0.07] | 6.23[3.81] | 8.03[3.10] |
| | LMHO | 0.21[0.04] | 0.43[0.06] | 7.07[2.14] | 8.39[3.28] |
| | LCOBS | 0.22[0.03] | 0.42[0.05] | 6.04[1.51] | 8.13[4.77] |
| | | Percentage difference with the P-value | | Difference with the P-value | |
| Walking speed | Offloading condition | Width (Control value, m; *difference from control, % and [P-value]*) | Length (Control value, m; *difference from control, % and [P-value]*) | Asymmetry of width (Control value, %; *difference from control, % and [P-value]*) | Asymmetry of length (Control value, %; *difference from control, % and [P-value]*) |
| Slow | Control | 0.21 | 0.31 | 5.90 | 11.33 |
| | SCO | 4[1.000] | 2[0.641] | 0.08[0.439] | 0.30[0.367] |
| | LCO | 7[0.537] | 7[0.062] | 0.86[0.543] | 0.80[0.731] |
| | SMHO | 5[0.625] | 3[0.272] | 0.30[0.266] | -0.52[0.748] |
| | LMHO | 6[0.534] | 4[0.155] | 0.50[0.534] | 2.38[0.266] |
| | LCOBS | 5[0.681] | 7[0.051] | 0.77[0.664] | 1.99[0.286] |
| Normal | Control | 0.22 | 0.35 | 4.81 | 9.15 |
| | SCO | 1[1.000] | 1[0.423] | -0.08[0.920] | -0.05[0.961] |
| | LCO | 5[0.204] | 1[0.560] | 0.26[0.590] | -1.77[0.105] |
| | SMHO | 1[0.695] | 6[0.049] | 0.89[0.087] | -0.12[0.911] |
| | LMHO | 1[0.746] | 4[0.018] | 1.54[0.019] | -0.24[0.880] |
| | LCOBS | -1[0.656] | 4[0.022] | -0.18[0.709] | -0.25[0.824] |
| Fast | Control | 0.21 | 0.41 | 5.08 | 7.18 |
| | SCO | -3[0.285] | 1[0.674] | 0.97[0.277] | 1.50[0.048] |
| | LCO | -1[0.649] | 5[0.022] | 1.55[0.252] | 1.12[0.364] |
| | SMHO | 0[0.853] | 3[0.378] | 1.30[0.331] | 0.75[0.411] |
| | LMHO | -1[0.329] | 6[0.014] | 2.10[0.022] | 1.39[0.223] |
| | LCOBS | 1[0.743] | 3[0.090] | 0.96[0.041] | 0.95[0.458] |

Percentage differences and P-values were calculated for the five offloading conditions and the control condition for 10 subjects at three walking speeds. P-values smaller than 0.05 were underlined. Control means control flat insole, SCO means small calcaneus offloading insole, LCO means larger calcaneus offloading insole, SMHO means small metatarsal head offloading insole, LMHO means large metatarsal head offloading insole, LCOBS means both sides large calcaneus offloading insole, and std means standard deviation.

**Table 4. The mean arch index and total contact area of five offloading conditions and the control condition for 10 subjects at three walking speeds.**

| Condition | Walking speed | Arch index (mean[std]) | Contact area (mm^2, mean[std]) |
|---|---|---|---|
| Control (Control flat insole) | Slow | 0.22[0.04] | 9387[1612] |
| | Normal | 0.21[0.06] | 9347[1756] |
| | Fast | 0.21[0.05] | 9489[1724] |
| SCO (Small calcaneus offloading insole) | Slow | 0.21[0.06] | 9004[1531] |
| | Normal | 0.21[0.06] | 9259[1781] |
| | Fast | 0.20[0.06] | 9212[1750] |
| LCO (Larger calcaneus offloading insole) | Slow | 0.22[0.05] | 9607[1607] |
| | Normal | 0.21[0.05] | 9545[1511] |
| | Fast | 0.21[0.05] | 9752[1745] |
| SMHO (Small metatarsal head offloading insole) | Slow | 0.20[0.05] | 9219[1513] |
| | Normal | 0.20[0.06] | 9407[1745] |
| | Fast | 0.20[0.05] | 9429[1557] |
| LMHO (Large metatarsal head offloading insole) | Slow | 0.21[0.04] | 9379[1451] |
| | Normal | 0.19[0.03] | 9258[1129] |
| | Fast | 0.19[0.05] | 9554[1401] |
| LCOBS (Both sides large calcaneus offloading insole) | Slow | 0.22[0.05] | 9412[1273] |
| | Normal | 0.21[0.04] | 9444[1312] |
| | Fast | 0.20[0.04] | 9573[1312] |
| Percentage difference with the P-value | | | |
| Condition | Walking speed | Arch index (Control value, no unit; *difference from control, % and [P-value]*) | Contact area (Control value, mm^2; *difference from control, % and [P-value]*) |
| Control (Control flat insole) | Slow | 0.22 | 9387 |
| | Normal | 0.21 | 9347 |
| | Fast | 0.21 | 9489 |
| SCO (Small calcaneus offloading insole) | Slow | -2%[0.900] | -4%[0.001] |
| | Normal | -2%[0.868] | -1%[0.556] |
| | Fast | -7%[0.034] | -3%[0.101] |
| LCO (Larger calcaneus offloading insole) | Slow | 3%[0.276] | 1%[0.526] |
| | Normal | 2%[1.000] | 1%[0.936] |
| | Fast | 1%[0.813] | 1%[0.379] |
| SMHO (Small metatarsal head offloading insole) | Slow | -7%[0.149] | -3%[0.174] |
| | Normal | -4%[0.262] | -1%[0.529] |
| | Fast | -9%[0.035] | -2%[0.215] |
| LMHO (Large metatarsal head offloading insole) | Slow | -9%[0.011] | -5%[0.031] |
| | Normal | -14%[0.035] | -6%[0.049] |
| | Fast | -14%[0.080] | -3%[0.136] |
| LCOBS (Both sides large calcaneus offloading insole) | Slow | 0%[0.279] | -1%[0.529] |
| | Normal | 1%[1.000] | 0%[0.726] |
| | Fast | -3%[0.303] | 0%[0.685] |

P-values were calculated for the five offloading conditions and the control condition for the 10 subjects at three walking speeds. P-values smaller than 0.05 were underlined.

et al [2] have developed a finite element model to optimize the offloading aperture shape to minimize the calcaneus load after offloading. In addition, offloading calcaneus region will increase the average peak plantar pressure and pressure time integrals in metatarsal head region, and vice versa. A possible way to solve this problem could be to modify the insole to

minimise pressure in the aperture region rather than just over the average peak plantar pressure for metatarsal head or calcaneus. A total contact insole [24] or a functionally graded insole [14] might be a better pressure management solution.

No difference is observed about the sagittal angles but the heel strike and toe-off velocity increase between 1% and 32% respectively when applying offloading insoles except SMHO, which is not reported in other papers to the authors' knowledge. The heel strike and toe-off velocity indicate a relationship with average peak plantar pressure when walking at low speeds, but the relationship is not clear when walking at normal and fast speeds. More parameters should be reviewed together with heel strike and toe-off velocity to obtain a robust relationship between kinematics and plantar pressure. It indicates the possibility of predicting plantar pressure according to the kinematic data which may be useful for real world DFU risk prediction. When offloading one region of the foot, the large offloading apertures always exhibit larger heel strike and toe-off velocity than the small offloading aperture, which indicates that larger offloading apertures have more effect on the kinematics.

The centre of force figure (Fig 4) can provide information on step width, length and asymmetry. At the same region, large single offloading insoles create more asymmetry of step width than small offloading insoles. In addition, symmetric large calcaneus offloading conditions can decrease the step width asymmetry by 0.1% to 0.61% compared with single side large calcaneus offloading condition, which indicates single offloading insoles will create asymmetric gait. Existing research reported that applying a twin shoe to orthopaedic shoes can improve the asymmetry but cannot totally eliminate the asymmetry problem (18). Although these changes are very small in the healthy population tested these may be larger in a diabetic population so it may be advisable to offload both feet to reduce the asymmetry. The change in width, length and asymmetry of step length are affected by the offloading insole although they do not show a clear relationship with offloading conditions on healthy subjects. Increased stance width is linked with poorer dynamic balance control [25]. Consequently, the centre of force figure is important because it can show the change in gait kinematics and kinetics from force plate data, which could be a useful support tool in the future to evaluate the diabetics' gait.

The limitations of this study are the shoe-mounted markers, the generic insole, small sample size and healthy subjects. In this study, the limitation is recognised of using shoe-mounted markers, as they cannot represent the foot's movement inside the shoe, unlike skin-mounted markers which measure actual foot motion [26]. However, we chose shoe-mounted markers to maintain the integrity of the footwear, considering its potential effect on gait. The measurement of foot motion will therefore have some inaccuracy, but the analysis of this study does not focus on foot motion, instead looks at ankle angle, therefore, the impact of using shoe-mounted markers is minimised. This study used generic insoles with mean offloading positions. The metatarsal head position showed more variation between subjects than the calcaneus region with some subjects showing average peak plantar pressure over $1^{st}$ metatarsal head and others over $3^{rd}$ metatarsal head which could be caused by pronation or supination walking conditions. Consequently, it is recommended to position the offloading aperture based on the measurement of subject specific peak plantar pressure. The results of Penny et al. [10] also observed the inter-subject variability of peak plantar pressure. Our sample size was small and chosen for convenience sampling, as is common for pilot studies of this type [20] therefore our results may not be statistically significant. However, the results of this study do show some indicative relationships between gait kinematics, and plantar pressure for offloading insole conditions but further work would be needed to determine if these relationships are significant. This study involved healthy participants to assess insole design safely and effect on gait kinematics and plantar pressure. This information could then be used in future studies on diabetic populations to ensure a safe and non-harmful approach. Although the current findings

do not directly apply to diabetic foot ulcer management or diabetic subjects, they are indicative of general gait changes which will help in guiding the development and evaluation of insoles that could significantly reduce the risk of diabetic foot ulceration. This effect (increasing heel strike and toe off speed) may be increased in a diabetic population which is likely to lead to balance problems in gait. In diabetic populations who have low gait speed and short stride length [27] reducing balance may have an adverse effect on activities of daily living therefore it is recommended that further investigations are taken into the balance and gait kinematics of diabetics whilst wearing offloading insoles.

## Conclusions

The application of offloading insoles can offload the region of interest; however, it will increase the plantar pressure around the offloading area creating an "edge effect". Some offloading insole configurations can increase heel strike and toe-off velocity, which could contribute to plantar pressure increases, but reduce the pressure time integral. Offloading insoles with larger offloading apertures create a superior offloading effect at the region of interest but this creates small increases in gait asymmetry in healthy subjects which may be larger in diabetic subjects. Further work is needed to understand the gait kinematic effects of offloading insoles in diabetics.

## Supporting information

**S1 File. Unprocessed data of plantar pressure, ground reaction force and gait kinematics.** This file can be accessed in Mendeley data (DOI: 10.17632/hfz543gjc7.1). Two folders are included in the 'S1_File' folder.
(DOCX)

**S1 Table. The slow, normal, and fast walking speed of 10 subjects.**
(DOCX)

## Author Contributions

**Conceptualization:** Helen Dawes, Frank L. Bowling, Neil D. Reeves, Andrew Weightman, Glen Cooper.

**Data curation:** Jiawei Shuang, Garry Massey, Maedeh Mansoubi.

**Formal analysis:** Jiawei Shuang, Athia Haron, Garry Massey.

**Funding acquisition:** Andrew Weightman.

**Investigation:** Jiawei Shuang, Garry Massey, Maedeh Mansoubi, Helen Dawes, Frank L. Bowling, Neil D. Reeves, Andrew Weightman, Glen Cooper.

**Methodology:** Jiawei Shuang, Athia Haron, Helen Dawes, Andrew Weightman, Glen Cooper.

**Project administration:** Jiawei Shuang, Glen Cooper.

**Resources:** Garry Massey, Maedeh Mansoubi, Helen Dawes.

**Software:** Jiawei Shuang, Garry Massey, Maedeh Mansoubi.

**Supervision:** Frank L. Bowling, Andrew Weightman, Glen Cooper.

**Visualization:** Jiawei Shuang, Glen Cooper.

**Writing – original draft:** Jiawei Shuang, Glen Cooper.

**Writing – review & editing:** Jiawei Shuang, Athia Haron, Maedeh Mansoubi, Helen Dawes, Frank L. Bowling, Neil D. Reeves, Andrew Weightman, Glen Cooper.

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
