## [Decision Letter · Decision Letter 0]

12 Jan 2024

PONE-D-23-30055The effect of calcaneus and metatarsal head offloading insoles on healthy subjects’ gait kinematics, kinetics, asymmetry, and the implications for plantar pressure management.PLOS ONE

Dear Dr. Shuang,

Thank you for submitting your manuscript to PLOS ONE. After careful consideration, we feel that it has merit but does not fully meet PLOS ONE’s publication criteria as it currently stands. Therefore, we invite you to submit a revised version of the manuscript that addresses the points raised during the review process.

We look forward to receiving your revised manuscript.

Kind regards,

Jianhong Zhou

Staff Editor

PLOS ONE

Journal Requirements:

"This work was partially funded by Engineering and Physical Sciences Research Council (EPSRC) grant number EP/W00366X/1.

 Helen Dawes and Maedeh Mansoubi are supported by NIHR Exeter BRC."

Please be informed that funding information should not appear in the Acknowledgments section or other areas of your manuscript. We will only publish funding information present in the Funding Statement section of the online submission form. 

"A. W. was the PI of Engineering and Physical Sciences Research Council (EPSRC

https://www.ukri.org/councils/epsrc/) grant number EP/W00366X/1.

The funders had no role in study design, data collection and analysis, decision to

publish, or preparation of the manuscript."

3. We notice that your supplementary figure (Figure 5) are uploaded with the file type 'Figure'. Please amend the file type to 'Supporting Information'. Please ensure that each Supporting Information file has a legend listed in the manuscript after the references list.

4. Please upload a copy of Supporting Information (S1 File, S2 Table) which you refer to in your text on page 21. 

Reviewers' comments:

Reviewer's Responses to Questions

**Comments to the Author**

1. Is the manuscript technically sound, and do the data support the conclusions?

Reviewer #1: Partly

Reviewer #2: No

2. Has the statistical analysis been performed appropriately and rigorously? 

Reviewer #1: No

Reviewer #2: No

3. Have the authors made all data underlying the findings in their manuscript fully available?

Reviewer #1: Yes

Reviewer #2: No

4. Is the manuscript presented in an intelligible fashion and written in standard English?

Reviewer #1: Yes

Reviewer #2: Yes

5. Review Comments to the Author

Reviewer #1: This work tested 6 types of insoles, including a control insole and off-loading insoles, on the treadmill walking. This study analyzed plantar pressure and walking kinematics. Even this study performed in healthy subjects but it aimed to used the results for diabetic patients. There are several concerns needed to be clarified.

1. Did the authors measure the arch index of the participants? Or any contact area is analyzed with the F-scan.

2. As this study used 3 measurement tools (F-scan, motion capture and force plate), how did the authors sync the signals from three measurement?

3. It is better to provide the pattern of gait cycles on the treadmill to demonstrate PTI.

4. Please provide the plantar pressure distribution from F-scan of each insole. These information will give clear picture of edge effect.

5. Why did the authors choose 10 cycles to represent in this study? How long did the participant walk on the treadmill?

6. There is a typo error on line 287 (page 12) for 10 subjects. It typed as 1o which is "o" character, not "0" number.

7. As the authors addressed about the limitations of this study, how can the finding be translated for the diabetic foot ulceration management? It is mandatory to do a further study in the diabetic patients as the authors mentioned but what the finding in healthy subjects can give to the aspect of the insole design.

Reviewer #2: Generally, the topic of this study look interesting and the outcome in this study is strong receiving from the laboratory. My major concern is the statistical analysis of this study, there is no clear information regarding this, which affect the result of this study. Any details in the discussion part should be described in the correlation with the result, the result part did not show the p-vale, anyways the authors use the word “significant” to explain the result. With regards to specific comments in each part, my comments are described following.

SESSION INTRODUCTION

Point 1 Authors: “Specially designed insoles can be part of an effective DFU prevention strategy which includes maintaining mobility and physical activity as part of a holistic management package [6,7].”

Reviewer: How about the other interventions? The authors should imply general interventions for DFU for a few sentences before raising the insole as the important intervention.

Point 2 Authors: “Lin et al. [8] evaluated the average peak plantar pressure in the forefoot region of an insole with removable plugs and a support arch using a pedar in-shoe plantar pressure measurement system on diabetic patients”

Reviewer: Please explain more about the effects of DFU on increasing forefoot plantar pressure before describing the results of Lin et al. study.

Point 3 Authors: “Previous research has clearly shown that offloading insoles are effective at reducing normal plantar pressure in a specific region, with some impact on surrounding areas”

“To the authors’ knowledge, the effect of offloading insoles on gait kinematics, kinetics and plantar pressure has not been investigated.

Reviewer: The authors have already explained about the edge effects of offloading insole. Please describe more in the research gap about how to solve this problem in the present study.

SESSION METHODS

Point 4 Authors: “Ten participants were recruited for this experiment,”

Reviewer: Why the present study conduct only 10 healthy participants? How about the sample size calculation?

Point 5 Authors: “Ten healthy participants (five male and five female) who did not have diabetes were recruited”

Reviewer: The author should clarify the exclusion criteria. How about the history of lower-extremity fracture, other systemic conditions, the musculoskeletal symptoms of lower back and lower extremity etc.?

Point 6 Authors: “Six types of offloading insoles were manufactured including a control flat insole (Control), a small calcaneus offloading insole (SCO), a large calcaneus offloading insole (LCO), a small metatarsal head offloading insole (SMHO), a large metatarsal head offloading insole (LMHO), and a large calcaneus offloading insole for both feet (LCOBS) (Fig 1C). The LCOBS was the symmetric condition to compare to unilateral insoles (LCO).”

Reviewer: Please explain why the authors use the symmetric condition for only the calcaneus offloading, not the metatarsal head offloading.

Point 7 Authors: “The 1st and 2nd metatarsal heads location was defined from the mean anatomical locations measured from a small sample of 6 healthy participants which gave L1, W1, L2, and W2 are 71%, 85%, 74% and 63% respectively,”

Reviewer: Why the authors use the mean from 6 healthy participants, not from 10 participants.

Point 8 Authors: “A total of 16 markers were affixed to the diabetic shoes, excluding the body markers (Fig 2).”

Reviewer: The author should explain more in the discussion part about the advantage and disadvantage of the shoe-mounted marker and the skin-mounted marker.

Point 9 Authors: “The data from the 10 participants were evaluated on an individual basis using the data from the individuals control insole condition as a baseline to compare kinematic and pressure differences.”

Reviewer: The author should clarify the statistical method that was used in this study. There is no clearly information about the method to compare the data among 6 conditions using the p-value.

SESSION RESULTS

Point 10 Authors: “Table 1 shows the mean average peak plantar pressure (APPP) and shank distal end velocity percentage difference between five offloading conditions and control condition at region of foot (RoF) and region of interest (RoI) (defined in Fig 3) of 10 subjects at three walking speeds.”

Reviewer: It would be better to describe the mean and sd of each outcome and re analyze the statistical difference with the p-value among the 6 conditions of insole.

Point 11 Authors: “Table 2 shows the mean pressure time integrals (PTI) percentage difference between five offloading conditions and control condition at RoF and RoI of 1o subjects at three walking speeds.”

Reviewer: It would be better to describe the mean and sd of each outcome and re analyze the statistical difference with the p-value among the 6 conditions of insole.

Point 12 Authors: “Table 3 shows the mean value of width of centre of force (CoF), length of CoF, and asymmetry difference between five offloading conditions and control condition of 10 subjects at three walking speeds.”

Reviewer: It would be better to describe the mean and sd of each outcome and re analyze the statistical difference with the p-value among the 6 conditions of insole.

SESSION DISCUSSION

Point 13 Authors: “A possible way to solve this problem could be to modify the insole to minimise pressure in the aperture region rather than just over the average peak plantar pressure for metatarsal head or calcaneus.”

Reviewer: The author should clarify how the present study design the insole to solve the problem from the previous studies.

Point 14 Authors: “No significant trend is found about the sagittal angles but the heel strike and toe-off velocity increase between 1% and 32% respectively when applying offloading insoles except SMHO, which is not reported in other papers to the authors’ knowledge.”

Reviewer: Please describe more detail about how to receive the significant value, since the author did not report the significant value in the table data.

Point 15 Authors: “Although as a pilot study, the result of this study is not statistically significant, it gives a general trend of relationship between gait kinematics, kinetics, asymmetry and plantar pressure.”

Reviewer: The author should add in the title about the word “Pilot study”. Again, please describe more detail about how to receive the significant value, since the author did not report the significant value in the table data.

6. PLOS authors have the option to publish the peer review history of their article (what does this mean?). If published, this will include your full peer review and any attached files.

Reviewer #1: **Yes: **Surapong Chatpun

Reviewer #2: No

---

## [Author Response · Author response to Decision Letter 0]

22 Feb 2024

Thank you for sharing your valuable feedback and comments. We have copied and pasted the contents of 'Response to Reviewers' in this message. For figures and tables, please refer to the 'Response to Reviewers' file updated in the PLOS ONE editorial manager.

Below is the copy of 'Response to Reviewers' file:

21 Feb 2024

Dear Dr Jianhong Zhou

Re: PLOS ONE Decision: Revision required [PONE-D-23-30055] - [EMID:db690a3d5d331f19]

Thank you for giving us the opportunity to submit a revised draft of our manuscript titled ‘The effect of calcaneus and metatarsal head offloading insoles on healthy subjects’ gait kinematics, kinetics, asymmetry, and the implications for plantar pressure management’ to PLOS ONE.

We appreciate the time and effort that you and the reviewers have dedicated to providing your valuable feedback and comments on our manuscript. We have tried to incorporate changes to reflect the suggestions provided by the reviewers.

Here is a point-by-point response to the reviewer’s comments and concerns. Lines quoted are for the ‘Revised Manuscript with Track Changes’. 

Comments from the Review #1:

Summary of Comments:

This work tested 6 types of insoles, including a control insole and off-loading insoles, on the treadmill walking. This study analyzed plantar pressure and walking kinematics. Even this study performed in healthy subjects but it aimed to used the results for diabetic patients. There are several concerns needed to be clarified.

Response:

Thank you for taking the time to read our manuscript. We really appreciate the feedback and 

insightful comments on our manuscript. We have tried to address your concerns on comments for diabetic patients in the responses to the comments below.

Comment 1:

Did the authors measure the arch index of the participants? Or any contact area is analyzed with the F-scan.

Response:

Thank you for highlighting this. We have now calculated the average arch index and total foot contact area of the six insole conditions for ten subjects at three walking speeds. We have now given the method detail in line 181 (page 8), showed the result in line 292 (page 14), included the table in line 298 (page 14) and in line 450 (page 20) of our manuscript, discussed the sample size limitation in line 353 (page 17).

See the highlighted sentences below for changes and the table:

Added to method section (line 181):

“The arch index was calculated from normal plantar pressure measurements taken with the F-scan system, using the method outlined in reference [24]. In addition, the total contact area of the foot was recorded and exported.”

Added to the Results section (line 292):

“Table 4 shows the mean arch index and total contact area. P-values were calculated for the five offloading conditions and the control condition for the 10 subjects at three walking speeds (S6 Table). There was a statistical trend that showed decreases in the arch index for the SCO condition during fast walking (-7%, P<0.05), the SMHO condition during fast walking (-9%, P<0.05), and the LMHO condition during slow and normal walking (-9%, P<0.05; -14%, P<0.05). A statistical trend showed decreases in the total contact area for the SCO condition during slow walking (-4%, P<0.05) and the LMHO condition during slow and normal walking (-5%, P<0.05; -6%, P<0.05).

Added Table 4 to the main text (line 298):

Table 4. The mean arch index and total contact area of five offloading conditions and the control condition for 10 subjects at three walking speeds.

Condition Walking speed Arch index (mean[std]) Contact area (mm^2, mean[std])

Control

(Control flat insole)

　 Slow 0.22[0.04] 9387[1612]

 Normal 0.21[0.06] 9347[1756]

 Fast 0.21[0.05] 9489[1724]

SCO

(Small calcaneus offloading insole) Slow 0.21[0.06] 9004[1531]

 Normal 0.21[0.06] 9259[1781]

 Fast 0.20[0.06] 9212[1750]

LCO

(Larger calcaneus offloading insole) Slow 0.22[0.05] 9607[1607]

 Normal 0.21[0.05] 9545[1511]

 Fast 0.21[0.05] 9752[1745]

SMHO

(Small metatarsal head offloading insole) Slow 0.20[0.05] 9219[1513]

 Normal 0.20[0.06] 9407[1745]

 Fast 0.20[0.05] 9429[1557]

LMHO

(Large metatarsal head offloading insole) Slow 0.21[0.04] 9379[1451]

 Normal 0.19[0.03] 9258[1129]

 Fast 0.19[0.05] 9554[1401]

LCOBS

(Both sides large calcaneus offloading insole) Slow 0.22[0.05] 9412[1273]

 Normal 0.21[0.04] 9444[1312]

 Fast 0.20[0.04] 9573[1312]

“

Added to the discussion section (line 353):

“Our sample size was small and chosen for convenience sampling, as is common for pilot studies of this type [20] therefore our results may not be statistically significant. However, the results of this study do show some indicative relationships between gait kinematics, and plantar pressure for offloading insole conditions but further work would be needed to determine if these relationships are significant.”

Added to the supporting information section (line 450):

“S6 Table. The mean arch index and total contact area along with the percentage difference and P-value between five offloading conditions and the control condition for 10 subjects at three walking speeds.

Percentage difference with the P-value 

Condition Walking speed Arch index (Control value, no unit;

difference from control, % and [P-value]) Contact area 

(Control value, mm^2;

difference from control, % and [P-value])

Control

(Control flat insole)

 Slow 0.22 9387

 Normal 0.21 9347

 Fast 0.21 9489

SCO

(Small calcaneus offloading insole) Slow -2%[0.90] -4%[<0.05]

 Normal -2%[0.87] -1%[0.56]

 Fast -7%[<0.05] -3%[0.10]

LCO

(Larger calcaneus offloading insole) Slow 3%[0.28] 1%[0.53]

 Normal 2%[1.00] 1%[0.94]

 Fast 1%[0.81] 1%[0.38]

SMHO

(Small metatarsal head offloading insole) Slow -7%[0.15] -3%[0.17] 

 Normal -4%[0.26] -1%[0.53]

 Fast -9%[<0.05] -2%[0.22]

LMHO

(Large metatarsal head offloading insole) Slow -9%[<0.05] -5%[<0.05]

 Normal -14%[<0.05] -6%[<0.05]

 Fast -14%[0.08] -3%[0.14]

LCOBS

(Both sides large calcaneus offloading insole) Slow 0%[0.28] -1%[0.53]

 Normal 1%[1.00] 0%[0.73]

 Fast -3%[0.30] 0%[0.69]

Comment 2:

As this study used 3 measurement tools (F-scan, motion capture and force plate), how did the authors sync the signals from three measurement?

Response:

Thank you for highlighting this. We achieved synchronization between the two force plates and the motion capture system through a hardwired connection. We used the initial heel strike as an event to synchronize the in-shoe pressure measurement system with both the force plates and the motion capture system. We added these details to line 148 (page 6) of our manuscript.

See the shaded sentences below for changes and the figure:

“…and F-Scan research (v7.00 -19), respectively. Synchronization between the two force plates and the motion capture system was achieved through a hardwired connection. The initial heel strike was utilised as a common event to synchronize the in-shoe pressure measurement system with both the force plates and the motion capture system.”

Comment 3:

It is better to provide the pattern of gait cycles on the treadmill to demonstrate PTI.

Response:

Thank you for your suggestions. We added the figure to provide this information in line 249 (page 11) and line 263 (page 12) of our manuscript.

See the shaded sentences below for changes and the figure:

“Fig 6 presents the plantar pressure and the pressure time integral in the regions of interest at the calcaneus and metatarsal heads during one gait cycle of ten subjects walking at normal speed. Fig 6 shows that the SCO, LCO, and LCOBS insoles decreased the peak plantar pressure in calcaneus region, while no difference of metatarsal heads peak plantar pressure was observed when utilising the SMHO and LMHO insoles. Additionally, all the conditions show differences in peak plantar pressure profiles across the gait cycle for either the calcaneus or metatarsal head region.

. Table 2 shows the mean…”

“Fig 6: The plantar pressure and the pressure time integral in the regions of interest at the calcaneus and metatarsal heads during one gait cycle of ten subjects walking at normal speed. The black circles and squares indicate the locations of peak plantar pressure at the calcaneus and metatarsal heads regions of interest respectively, while the blue and red shadows represent the pressure time integral at the calcaneus and metatarsal heads regions of interest, respectively.” 

Comment 4:

Please provide the plantar pressure distribution from F-scan of each insole. These information will give clear picture of edge effect.

Response:

Thank you for your suggestions. We add the figure of the plantar pressure distribution from F-scan in line 225 (page 10) and line 252 (page 10) of our manuscript.

See the shaded sentences below for changes and the figure:

“Fig 5 presents the average peak plantar pressure (APPP) of one subject under five offloading conditions and a control condition, each recorded at a normal walking speed. Table 1 shows…”

“Fig 5. The average peak plantar pressure of one subject under five offloading conditions and a control condition, each recorded at a normal walking speed. The red dash cycle is the location of offloading aperture.”

Comment 5:

Why did the authors choose 10 cycles to represent in this study? How long did the participant walk on the treadmill?

Response:

Thank you for pointing this out. Participants walked for 10 minutes in total on the treadmill for three speeds of each condition. For each walking speed, 20 seconds was used to adjust the treadmill speed and the remaining three minutes were used to walk at constant speed. The ten gait cycles were taken from the middle portion of the three-minute constant speed walking. From observation of the data variability within this period was extremely low. Other researchers have typically used ten gait cycles for their analysis, see ref Shi et al. (2022) and Moe-Nilssen, R (1998). Furthermore, we added a better explanation on this within the text in line 159 (page 7) of our manuscript. 

See the shaded sentences below for changes: 

“normal walking speed (NWS) which was their self-selected walking speed to reflect their preferred walking pace. Subjects then walked on the treadmill at a slow speed (NWS - 0.2 m/s), then normal speed (NWS) and finally fast speed (NWS + 0.2 m/s) for 3 minutes and 20 seconds each (total duration 10 minutes). The participants started from rest on the treadmill and the speed was changed through normal, fast and slow walking speeds.”

References:

Shi, Q.Q., Li, P.L., Yick, K.L., Li, N.W. and Jiao, J., 2022. Effects of contoured insoles with different materials on plantar pressure offloading in diabetic elderly during gait. Scientific Reports, 12(1), p.15395.

Moe-Nilssen, R., 1998. Test-retest reliability of trunk accelerometry during standing and walking. Archives of physical medicine and rehabilitation, 79(11), pp.1377-1385.

Comment 6:

There is a typo error on line 263 (page 12) for 10 subjects. It typed as 1o which is "o" character, not "0" number.

Response:

Thank you for pointing this out. We correct this typo in line 267 (page 12) of our manuscript.

See the shaded sentences below for changes:

“Table 2. The mean pressure time integrals and shank distal end velocity of five offloading conditions and the control condition at the region of foot (RoF) and region of interest (RoI) for 10 subjects at three walking speeds.”

Comment 7:

As the authors addressed about the limitations of this study, how can the finding be translated for the diabetic foot ulceration management? It is mandatory to do a further study in the diabetic patients as the authors mentioned but what the finding in healthy subjects can give to the aspect of the insole design.

Response:

Thank you we agree that there is a need for further study in diabetic subjects. This study was initially conducted on healthy participants before diabetic participants, to allow for the refinement of insole designs without risking harm to diabetic participants. The behaviours observed in healthy participants can offer indications of how diabetic participants might respond and provide control data for comparisons and better understanding of gait metrics either causing or resulting from diabetes. This is the first study to jointly measure gait kinematics and plantar pressure changes caused by offloading insoles, which remove insole material from high-risk regions. Some offloading insole configurations may increase heel strike and toe-off velocity, potentially leading to increases in plantar pressure. Although the current findings do not directly apply to diabetic foot ulcer management or diabetic subjects, they are indicative of general gait changes which will help in guiding the development and evaluation of insoles that could significantly reduce the risk of diabetic foot ulceration.

Furthermore, we added a better explanation on this within the text in line 357 (page 17) of our manuscript. 

See the shaded sentences below for changes: 

“results of Penny et al. [10] also observed the inter-subject variability of peak plantar pressure. This study involved healthy participants to assess insole design safety and effect on gait kinematics and plantar pressure. This information could then be used in future studies on diabetic populations to ensure a safe and non-harmful approach. Although the current findings do not directly apply to diabetic foot ulcer management or diabetic subjects, they are indicative of general gait changes which will help in guiding the development and evaluation of insoles that could significantly reduce the risk of diabetic foot ulceration. This effect (increasing heel strike and toe off speed) may be increased in a diabetic population which is likely to lead to balance problems in gait.”

Comments from the Review #2:

Summary of Comments:

Generally, the topic of this study look interesting and the outcome in this study is strong receiving from the laboratory. My major concern is the statistical analysis of this study, there is no clear information regarding this, which affect the result of this study. Any details in the discussion part should be described in the correlation with the result, the result part did not show the p-vale, anyways the authors use the word “significant” to explain the result. With regards to specific comments in each part, my comments are described following.

Response:

Thank you for this insightful comment. We agree and more detailed information on the statistical analysis has been added to the paper.

Comment 1:

INTRODUCTION: “Specially designed insoles can be part of an effective DFU prevention strategy which includes maintaining mobility and physical activity as part of a holistic management package [6,7].”

Reviewer: How about the other interventions? The authors should imply general interventions for DFU for a few sentences before raising the insole as the important intervention.

Response:

Thank you for pointing this out. We add a few sentences to introduce the general interventions for DFU in line 41 (page 2) of our manuscript. 

See the shaded sentences below for changes: 

“General interventions for diabetic foot ulcer (DFU) include self-care practices, education and self-management, the employment of footwear and orthotic devices, along with diverse clinical treatment strategies [6]. Among these, the use of appropriate footwear stands out as a critical element in the prevention of DFU [6]. Specially designed insoles…”

Comment 2:

INTRODUCTION: “Lin et al. [8] evaluated the average peak plantar pressure in the forefoot region of an insole with removable plugs and a support arch using a pedar in-shoe plantar pressure measurement system on diabetic patients”

Reviewer: Please explain more about the effects of DFU on increasing forefoot plantar pressure before describing the results of Lin et al. study.

Response:

Thank you for your suggestion. We add a sentence to discuss the relationship between the increasing forefoot pressure and the DFU in line 45 (page 2) of our manuscript. 

See the shaded sentences below for changes: 

” Significantly higher peak forefoot pressures are observed in diabetic subjects, for example researchers report 608 kPa and 373 kPa in diabetics with severe and moderate neu

---

## [Decision Letter · Decision Letter 1]

27 Mar 2024

PONE-D-23-30055R1The effect of calcaneus and metatarsal head offloading insoles on healthy subjects’ gait kinematics, kinetics, asymmetry, and the implications for plantar pressure management.PLOS ONE

Dear Dr. Shuang,

Thank you for submitting your manuscript to PLOS ONE. After careful consideration, we feel that it has merit but does not fully meet PLOS ONE’s publication criteria as it currently stands. Therefore, we invite you to submit a revised version of the manuscript that addresses the points raised during the review process.

We look forward to receiving your revised manuscript.

Kind regards,

Surapong chatpun, Ph.D.

Guest Editor

PLOS ONE

Additional Editor Comments:

There is still a major concern from the reviewer. Therefore, we ask you to address and response to the Reviewer#2's comment.

Reviewer#2:

My major concern is still the statistical analysis of this study, the author should clearly present the real number of p-value in the table, the writing only P < 0.05 within the text is not enough. Please re-consider to present the p-value in the table other than only the mean and standard deviation and rewrite the text within the paragraph especially the variables without significant and the variables with significance.

Reviewers' comments:

Reviewer's Responses to Questions

**Comments to the Author**

1. If the authors have adequately addressed your comments raised in a previous round of review and you feel that this manuscript is now acceptable for publication, you may indicate that here to bypass the “Comments to the Author” section, enter your conflict of interest statement in the “Confidential to Editor” section, and submit your "Accept" recommendation.

Reviewer #1: All comments have been addressed

Reviewer #2: All comments have been addressed

2. Is the manuscript technically sound, and do the data support the conclusions?

Reviewer #1: Yes

Reviewer #2: Yes

3. Has the statistical analysis been performed appropriately and rigorously? 

Reviewer #1: Yes

Reviewer #2: No

4. Have the authors made all data underlying the findings in their manuscript fully available?

Reviewer #1: Yes

Reviewer #2: Yes

5. Is the manuscript presented in an intelligible fashion and written in standard English?

Reviewer #1: Yes

Reviewer #2: Yes

6. Review Comments to the Author

Reviewer #1: The authors addressed the responses to the comments completely and clearly. The authors added more information to make the content better than the previous version.

Reviewer #2: My major concern is still the statistical analysis of this study, the author should clearly present the real number of p-value in the table, the writing only P < 0.05 within the text is not enough. Please re-consider to present the p-value in the table other than only the mean and standard deviation and rewrite the text within the paragraph especially the variables without significant and the variables with significance.

7. PLOS authors have the option to publish the peer review history of their article (what does this mean?). If published, this will include your full peer review and any attached files.

Reviewer #1: No

Reviewer #2: No

---

## [Author Response · Author response to Decision Letter 1]

10 Apr 2024

Please see the uploaded file 'Response to Reviewers' to view the tables.

10th April 2024

Dear Dr Surapong Chatpun,

Re: PLOS ONE Decision: Revision required [PONE-D-23-30055] - [EMID:db690a3d5d331f19]

Thank you for giving us the opportunity to submit a revised draft of our manuscript titled ‘The effect of calcaneus and metatarsal head offloading insoles on healthy subjects’ gait kinematics, kinetics, asymmetry, and the implications for plantar pressure management’ to PLOS ONE.

We appreciate the time and effort that you and the reviewers have dedicated to providing your valuable feedback and comments on our manuscript. We have tried to incorporate changes to reflect the suggestions provided by the reviewers.

Here is the response to the reviewer’s comments and concerns. Lines quoted are for the ‘Revised Manuscript with Track Changes’. 

Comments from the Review #1:

Review Comments:

The authors addressed the responses to the comments completely and clearly. The authors added more information to make the content better than the previous version.

Response:

Thank you for your positive feedback. We're glad to hear that the additional information and clarifications have enhanced the content and addressed your comments effectively. If there are any more questions or further input, we'd be happy to hear them.

Comments from the Review #2:

Review Comments:

My major concern is still the statistical analysis of this study, the author should clearly present the real number of p-value in the table, the writing only P < 0.05 within the text is not enough. Please re-consider to present the p-value in the table other than only the mean and standard deviation and rewrite the text within the paragraph especially the variables without significant and the variables with significance.

Response:

Thank you for your comment. We appreciate your desire to include all the P-values within the main text. We have now moved this information from the supplementary information to the main text and made the following modifications:

• Added P-values to table 1-4.

• Added P-values to the text descriptions explaining if they were significant or insignificant.

Full details of this are shown below in the “Changes Section” and in the amended manuscript attached.

Thank you again for your time and consideration of our research paper.

Kind regards

Jiawei Shuang

Changes Section

Please see the tables below for changes associated with P-value real number:

Table 1. The mean average peak plantar pressure and shank distal end velocity of five offloading conditions and the control condition at the region of foot (RoF) and region of interest (RoI) for 10 subjects at three walking speeds. 

Walking speed Offloading condition Velocity (m/s, mean[std]) APPP at region of foot (kPa, mean[std]) APPP at region of interest (kPa, mean[std])

 Heel strike Toe-off Calcaneus Metatarsal head Toe Calcaneus Metatarsal head Toe

Slow Control 0.60[0.08] 0.96[0.09] 469[107] 530[166] 512[236] 469[107] 530[166] 512[236]

 SCO 0.67[0.10] 1.06[0.08] 495[115] 570[241] 482[206] 367[164] 490[235] 380[254]

 LCO 0.79[0.15] 1.11[0.06] 463[157] 672[340] 499[241] 164[104] 586[403] 367[291]

 SMHO 0.57[0.06] 0.95[0.12] 512[115] 562[151] 535[218] 405[97] 437[189] 346[249]

 LMHO 0.68[0.08] 1.09[0.12] 522[121] 573[152] 546[221] 428[88] 398[231] 352[245]

 LCOBS 0.79[0.14] 1.10[0.08] 426[110] 615[223] 466[206] 176[112] 556[258] 331[239]

Normal Control 0.61[0.11] 0.96[0.12] 490[117] 535[165] 681[331] 490[117] 535[165] 681[331]

 SCO 0.69[0.10] 1.09[0.07] 551[134] 559[175] 613[209] 405[196] 492[110] 432[219]

 LCO 0.79[0.12] 1.11[0.09] 566[178] 654[201] 585[247] 146[44] 602[180] 413[288]

 SMHO 0.60[0.08] 0.98[0.08] 553[143] 579[146] 611[254] 474[128] 469[199] 385[292]

 LMHO 0.70[0.16] 1.05[0.06] 557[154] 722[289] 691[271] 454[129] 484[260] 418[266]

 LCOBS 0.74[0.11] 1.09[0.10] 541[174] 724[299] 627[240] 149[65] 652[326] 350[322]

Fast Control 0.65[0.11] 0.97[0.10] 522[113] 596[228] 671[253] 522[113] 596[228] 671[253]

 SCO 0.73[0.17] 1.12[0.10] 608[131] 639[219] 718[292] 449[207] 571[170] 477[299]

 LCO 0.75[0.13] 1.10[0.06] 571[215] 646[195] 687[285] 213[75] 584[209] 493[350]

 SMHO 0.62[0.11] 0.98[0.13] 603[145] 652[155] 693[256] 535[156] 519[179] 477[294]

 LMHO 0.66[0.09] 1.06[0.05] 616[127] 745[265] 715[225] 534[123] 453[300] 512[308]

 LCOBS 0.80[0.11] 1.14[0.07] 501[99] 777[330] 676[242] 222[94] 715[357] 320[187]

Walking speed Offloading condition Velocity

(Control value, m/s;

difference from control, % and [P-value]) APPP at region of foot

(Control value, kPa;

difference from control, % and [P-value]) APPP at region of interest

(Control value, kPa;

difference from control, % and [P-value])

 Heel strike Toe-off Calcaneus Metatarsal head Toe Calcaneus Metatarsal head Toe

Slow Control 0.6 0.96 469 530 512 469 530 512

 SCO 12[0.091] 11[0.000] 6[0.175] 4[0.234] 9[0.673] -23[0.040] -11[0.426] -14[0.148]

 LCO 30[0.003] 17[0.001] -1[0.892] 21[0.049] 14[0.752] -62[0.001] -1[0.521] -14[0.188]

 SMHO -6[0.117] -2[0.598] 11[0.189] 8[0.265] 24[0.730] -13[0.006] -12[0.294] -8[0.157]

 LMHO 15[0.020] 15[0.044] 13[0.096] 10[0.240] 24[0.665] -7[0.121] -21[0.195] -10[0.170]

 LCOBS 29[0.004] 17[0.010] -7[0.277] 16[0.031] 6[0.535] -59[0.001] 2[0.581] -18[0.104]

Normal Control 0.61 0.96 490 535 681 490 535 681

 SCO 15[0.005] 16[0.001] 13[0.002] 6[0.192] -2[0.327] -19[0.108] -5[0.104] -31[0.030]

 LCO 32[0.002] 17[0.004] 17[0.142] 26[0.014] -2[0.379] -68[0.001] 16[0.039] -29[0.065]

 SMHO 1[0.909] 3[0.398] 15[0.149] 12[0.158] 0[0.256] -3[0.580] -7[0.429] -32[0.067]

 LMHO 15[0.004] 11[0.006] 14[0.091] 34[0.028] 21[0.809] -8[0.192] -7[0.579] -18[0.113]

 LCOBS 25[0.008] 15[0.013] 14[0.423] 36[0.015] 10[0.668] -67[0.001] 18[0.11] -33[0.073]

Fast Control 0.65 0.97 522 596 671 522 596 671

 SCO 12[0.006] 16[0.004] 17[0.001] 9[0.146] 9[0.366] -14[0.264] 0[0.674] -25[0.085]

 LCO 23[0.026] 16[0.005] 7[0.485] 22[0.025] 3[0.851] -57[0.001] 9[0.319] -30[0.083]

 SMHO -1[0.755] 2[0.426] 17[0.038] 14[0.240] 9[0.962] 2[0.666] -3[0.488] -22[0.093]

 LMHO 1[0.945] 11[0.023] 19[0.004] 30[0.164] 20[0.673] 3[0.552] -16[0.291] -10[0.270]

 LCOBS 24[0.001] 19[0.002] 0[0.661] 30[0.002] 13[0.967] -55[0.001] 18[0.072] -40[0.020]

Percentage differences and P-values were calculated for the five offloading conditions and the control condition for 10 subjects at three walking speeds. P-values smaller than 0.05 were underlined. APPP means average peak plantar pressure, SCO means small calcaneus offloading insole, LCO means larger calcaneus offloading insole, SMHO means small metatarsal head offloading insole, LMHO means large metatarsal head offloading insole, LCOBS means both sides large calcaneus offloading insole and std means standard deviation. 

Table 2. The mean pressure time integrals and shank distal end velocity of five offloading conditions and the control condition at the region of foot (RoF) and region of interest (RoI) for 10 subjects at three walking speeds. 

Walking speed Offloading condition Velocity (m/s, mean[std]) PTI at region of foot (kPa·s, mean[std]) PTI at region of interest (kPa·s, mean[std])

　 　 Heel strike Toe-off Calcaneus Metatarsal head Toe Calcaneus Metatarsal head Toe

Slow Control 0.60[0.08] 0.96[0.09] 1726[293] 1587[538] 1009[668] 1726[293] 1587[538] 1009[668]

　 SCO 0.67[0.10] 1.06[0.08] 1781[340] 1705[821] 917[497] 1171[533] 1395[599] 656[524]

　 LCO 0.79[0.15] 1.11[0.06] 1628[534] 1885[1121] 897[495] 406[179] 1627[1323] 528[475]

　 SMHO 0.57[0.06] 0.95[0.12] 1938[310] 1579[434] 978[549] 1548[306] 1109[469] 425[427]

　 LMHO 0.68[0.08] 1.09[0.12] 1915[348] 1542[480] 1007[633] 1622[266] 1024[733] 533[486]

　 LCOBS 0.79[0.14] 1.10[0.08] 1547[400] 1757[791] 843[586] 506[298] 1483[797] 428[316]

Normal Control 0.61[0.11] 0.96[0.12] 1457[197] 1489[596] 1402[982] 1457[197] 1489[596] 1402[982]

　 SCO 0.69[0.10] 1.09[0.07] 1648[359] 1488[667] 1090[618] 1285[563] 1182[368] 574[326]

　 LCO 0.79[0.12] 1.11[0.09] 1578[433] 1553[448] 996[493] 475[285] 1233[657] 527[408]

　 SMHO 0.60[0.08] 0.98[0.08] 1659[289] 1501[524] 883[298] 1197[471] 1100[737] 440[336]

　 LMHO 0.70[0.16] 1.05[0.06] 1574[368] 1508[534] 1071[507] 1145[442] 983[683] 580[510]

　 LCOBS 0.74[0.11] 1.09[0.10] 1513[501] 1850[960] 866[319] 371[301] 1585[954] 430[341]

Fast Control 0.65[0.11] 0.97[0.10] 1352[217] 1443[637] 1094[501] 1352[217] 1443[637] 1094[501]

　 SCO 0.73[0.17] 1.12[0.10] 1530[305] 1537[665] 982[346] 1147[476] 1259[478] 592[499]

　 LCO 0.75[0.13] 1.10[0.06] 1356[435] 1335[462] 903[462] 412[240] 1127[439] 622[416]

　 SMHO 0.62[0.11] 0.98[0.13] 1530[243] 1417[384] 971[330] 1210[235] 1131[456] 542[287]

　 LMHO 0.66[0.09] 1.06[0.05] 1624[192] 1449[315] 1045[399] 1306[172] 934[699] 687[462]

　 LCOBS 0.80[0.11] 1.14[0.07] 1116[226] 1691[723] 978[471] 475[326] 1418[808] 430[347]

Walking speed Offloading condition Velocity 

(Control value, m/s;

difference from control, % and [P-value]) PTI at region of foot

(Control value, kPa·s;

difference from control, % and [P-value]) PTI at region of interest

(Control value, kPa·s;

difference from control, % and [P-value])

　 　 Heel strike Toe-off Calcaneus Metatarsal head Toe Calcaneus Metatarsal head Toe

Slow Control 0.6 0.96 1726 1587 1009 1726 1587 1009

　 SCO 12[0.091] 11[0.000] 4[0.581] 4[0.679] 15[0.480] -29[0.025] -12[0.345] -23[0.442] 

　 LCO 30[0.003] 17[0.001] -6[0.437] 15[0.489] 16[0.314] -75[0.000] -4[0.347] -29[0.532] 

　 SMHO -6[0.117] -2[0.598] 14[0.061] 3[0.899] 27[0.354] -9[0.082] -20[0.218] -33[0.394] 

　 LMHO 15[0.020] 15[0.044] 12[0.104] 3[0.952] 26[0.338] -3[0.379] -26[0.259] -23[0.743] 

　 LCOBS 29[0.004] 17[0.010] -9[0.151] 15[0.616] 0[0.867] -68[0.000] -4[0.471] -31[0.291] 

Normal Control 0.61 0.96 1457 1489 1402 1457 1489 1402

　 SCO 15[0.005] 16[0.001] 12[0.039] -0[0.993] -7[0.487] -12[0.321] -15[0.046] -45[0.040] 

　 LCO 32[0.002] 17[0.004] 7[0.301] 11[0.540] -5[0.682] -65[0.000] -17[0.111] -41[0.074] 

　 SMHO 1[0.909] 3[0.398] 14[0.055] 8[0.950] -12[0.460] -19[0.054] -18[0.232] -50[0.039] 

　 LMHO 15[0.004] 11[0.006] 8[0.278] 6[0.916] 12[0.970] -22[0.032] -32[0.091] -29[0.141] 

　 LCOBS 25[0.008] 15[0.013] 4[0.731] 28[0.158] -11[0.469] -73[0.000] 4[0.634] -49[0.042] 

Fast Control 0.65 0.97 1352 1443 1094 1352 1443 1094

　 SCO 12[0.006] 16[0.004] 14[0.051] 7[NA] 1[0.357] -16[0.185] -8[0.970] -42[0.017] 

　 LCO 23[0.026] 16[0.005] -1[0.925] 6[NA] -11[0.222] -69[0.389] -10[0.742] -33[0.369] 

　 SMHO -1[0.755] 2[0.426] 15[0.063] 8[NA] 8[0.506] -9[0.081] -7[0.113] -34[0.033] 

　 LMHO 1[0.945] 11[0.023] 22[0.002] 12[NA] 20[0.825] -1[0.537] -27[0.738] -16[0.138] 

　 LCOBS 24[0.001] 19[0.002] -13[0.078] 24[NA] 8[0.598] -65[0.000] -1[0.864] -45[0.017] 

Percentage differences and P-values were calculated for the five offloading conditions and the control condition for 10 subjects at three walking speeds. P-values smaller than 0.05 were underlined. PTI means pressure time integral, SCO means small calcaneus offloading insole, LCO means larger calcaneus offloading insole, SMHO means small metatarsal head offloading insole, LMHO means large metatarsal head offloading insole, LCOBS means both sides large calcaneus offloading insole, and std means standard deviation.

Table 3. The mean centre of force width and length, and asymmetry of width and length of five offloading conditions and the control condition for 10 subjects at three walking speeds.

Mean and standard deviation data

Walking speed Offloading condition Width (m, mean[std]) Length (m, mean[std]) Asymmetry of width (mean[std]) Asymmetry of length (mean[std])

Slow Control 0.21[0.04] 0.31[0.05] 5.90[4.52] 11.33[4.70]

　 SCO 0.21[0.03] 0.32[0.04] 4.60[0.68] 13.70[7.27]

　 LCO 0.22[0.04] 0.33[0.05] 5.41[2.11] 12.29[4.16]

　 SMHO 0.21[0.04] 0.33[0.05] 4.75[2.10] 11.07[4.23]

　 LMHO 0.21[0.04] 0.33[0.06] 4.85[1.58] 13.06[5.05]

　 LCOBS 0.21[0.04] 0.33[0.05] 5.42[2.09] 12.69[2.74]

Normal Control 0.22[0.04] 0.35[0.06] 4.81[1.26] 9.15[2.60]

　 SCO 0.22[0.04] 0.36[0.06] 4.84[1.66] 9.10[3.55]

　 LCO 0.22[0.04] 0.35[0.06] 5.07[2.08] 7.38[2.23]

　 SMHO 0.22[0.03] 0.37[0.05] 5.70[2.10] 9.03[3.21]

　 LMHO 0.22[0.03] 0.37[0.06] 6.35[2.34] 8.90[4.65]

　 LCOBS 0.21[0.04] 0.36[0.06] 4.63[1.27] 8.90[3.14]

Fast Control 0.21[0.03] 0.41[0.06] 5.08[1.65] 7.18[1.91]

　 SCO 0.21[0.02] 0.41[0.06] 5.78[1.84] 8.69[2.44]

　 LCO 0.22[0.03] 0.42[0.06] 6.96[4.32] 8.33[4.61]

　 SMHO 0.21[0.03] 0.42[0.07] 6.23[3.81] 8.03[3.10]

　 LMHO 0.21[0.04] 0.43[0.06] 7.07[2.14] 8.39[3.28]

　 LCOBS 0.22[0.03] 0.42[0.05] 6.04[1.51] 8.13[4.77]

　 　 Percentage difference with the P-value Difference with the P-value

Walking speed Offloading condition Width (Control value, m;

difference from control, % and [P-value]) Length (Control value, m;

difference from control, % and [P-value]) Asymmetry of width (Control value, %;

difference from control, % and [P-value]) Asymmetry of length

(Control value, %;

difference from control, % and [P-value])

Slow Control 0.21 0.31 5.90 11.33

　 SCO 4[1.000] 2[0.641] 0.08[0.439] 0.30[0.367]

　 LCO 7[0.537] 7[0.062] 0.86[0.543] 0.80[0.731]

　 SMHO 5[0.625] 3[0.272] 0.30[0.266] -0.52[0.748]

　 LMHO 6[0.534] 4[0.155] 0.50[0.534] 2.38[0.266]

　 LCOBS 5[0.681] 7[0.051] 0.77[0.664] 1.99[0.286]

Normal Control 0.22 0.35 4.81 9.15

　 SCO 1[1.000] 1[0.423] -0.08[0.920] -0.05[0.961]

　 LCO 5[0.204] 1[0.560] 0.26[0.590] -1.77[0.105]

　 SMHO 1[0.695] 6[0.049] 0.89[0.087] -0.12[0.911]

　 LMHO 1[0.746] 4[0.018] 1.54[0.019] -0.24[0.880]

　 LCOBS -1[0.656] 4[0.022] -0.18[0.709] -0.25[0.824]

Fast Control 0.21 0.41 5.08 7.18

　 SCO -3[0.285] 1[0.674] 0.97[0.277] 1.50[0.048]

　 LCO -1[0.649] 5[0.022] 1.55[0.252] 1.12[0.364]

　 SMHO 0[0.853] 3[0.378] 1.30[0.331] 0.75[0.411]

　 LMHO -1[0.329] 6[0.014] 2.10[0.022] 1.39[0.223]

　 LCOBS 1[0.743] 3[0.090] 0.96[0.041] 0.95[0.458]

Percentage differences and P-values were calculated for the five offloading conditions and the control condition for 10 subjects at three walking speeds. P-values smaller than 0.05 were underlined. Control means control flat insole, SCO means small calcaneus offloading insole, LCO means larger calcaneus offloading insole, SMHO means small metatarsal head offloading insole, LMHO means large metatarsal head offloading insole, LCOBS means both sides large calcaneus offloading insole, and std means standard deviation. 

Table 4. The mean arch index and total contact area of five offloading conditions and the control condition for 10 subjects at three walking speeds.

Condition Walking speed Arch index (mean[std]) Contact area (mm^2, mean[std])

Control

(Control flat insole)

　 Slow 0.22[0.04] 9387[1612]

 Normal 0.21[0.06] 9347[1756]

 Fast 0.21[0.05] 9489[1724]

SCO

(Small calcaneus offloading insole) Slow 0.21[0.06] 9004[1531]

 Normal 0.21[0.06] 9259[1781]

 Fast 0.20[0.06] 9212[1750]

LCO

(Larger calcaneus offloading insole) Slow 0.22[0.05] 9607[1607]

 Normal 0.21[0.05] 9545[1511]

 Fast 0.21[0.05] 9752[1745]

SMHO

(Small metatarsal head offloading insole) Slow 0.20[0.05] 9219[1513]

 Normal 0.20[0.06] 9407[1745]

 Fast 0.20[0.05] 9429[1557]

LMHO

(Large metatarsal head offloading insole) Slow 0.21[0.04] 9379[1451]

 Normal 0.19[0.03] 9258[1129]

 Fast 0.19[0.05] 9554[1401]

LCOBS

(Both sides large calcaneus offloading insole) Slow 0.22[0.05] 9412[1273]

 Normal 0.21[0.04] 9444[1312]

 Fast 0.20[0.04] 9573[1312]

Percentage difference with the P-value 

Condition Walking speed Arch index (Control value, no unit;

difference from control, % and [P-value]) Contact area 

(Control value, mm^2;

difference from control, % and [P-value])

Control

(Control flat insole)

 Slow 0.22 9387

 Normal 0.21 9347

 Fast 0.21 9489

SCO

(Small calcaneus offloading insole) Slow -2%[0.900] -4%[0.001]

 Normal -2%[0.868] -1%[0.556]

 Fast -7%[0.034] -3%[0.101]

LCO

(Larger calcaneus offloading insole) Slow 3%[0.276] 1%[0.526]

 Normal 2%[1.000] 1%[0.936]

 Fast 1%[0.813] 1%[0.379]

SMHO

(Small metatarsal head offloading insole) Slow -7%[0.149] -3%[0.174] 

 Normal -4%[0.262] -1%[0.529]

 Fast -9%[0.035] -2%[0.215]

LMHO

(Large metatarsal head offloading in

---

## [Decision Letter · Decision Letter 2]

2 May 2024

The effect of calcaneus and metatarsal head offloading insoles on healthy subjects’ gait kinematics, kinetics, asymmetry, and the implications for plantar pressure management.

PONE-D-23-30055R2

Dear Dr. Shuang,

We’re pleased to inform you that your manuscript has been judged scientifically suitable for publication and will be formally accepted for publication once it meets all outstanding technical requirements.

Kind regards,

Surapong chatpun, Ph.D.

Guest Editor

PLOS ONE

Additional Editor Comments (optional):

-NO-

Reviewers' comments:

Reviewer's Responses to Questions

**Comments to the Author**

1. If the authors have adequately addressed your comments raised in a previous round of review and you feel that this manuscript is now acceptable for publication, you may indicate that here to bypass the “Comments to the Author” section, enter your conflict of interest statement in the “Confidential to Editor” section, and submit your "Accept" recommendation.

Reviewer #2: All comments have been addressed

2. Is the manuscript technically sound, and do the data support the conclusions?

Reviewer #2: Yes

3. Has the statistical analysis been performed appropriately and rigorously? 

Reviewer #2: Yes

4. Have the authors made all data underlying the findings in their manuscript fully available?

Reviewer #2: Yes

5. Is the manuscript presented in an intelligible fashion and written in standard English?

Reviewer #2: Yes

6. Review Comments to the Author

Reviewer #2: The author provide all correction. I agree to accept and publish. The language is ok and already adapted.

7. PLOS authors have the option to publish the peer review history of their article (what does this mean?). If published, this will include your full peer review and any attached files.

Reviewer #2: No

---

## [Editor Report · Acceptance letter]

8 May 2024

PONE-D-23-30055R2 

PLOS ONE

Dear Dr. Shuang, 

I'm pleased to inform you that your manuscript has been deemed suitable for publication in PLOS ONE. Congratulations! Your manuscript is now being handed over to our production team.

Kind regards, 

on behalf of

Professor Surapong chatpun 

Guest Editor

PLOS ONE